
# LISFLOOD-FP 8.2: GPU-accelerated multiwavelet discontinuous Galerkin solver with dynamic resolution adaptivity for rapid, multiscale flood simulation

Alovya Ahmed Chowdhury[1], Georges Kesserwani[1]

Department of Civil and Structural Engineering, University of Sheffield, Sheffield, S10 2TN, United Kingdom

**Correspondence:** Alovya Chowdhury (alovya.chowdhury@gmail.com) and Georges Kesserwani
(g.kesserwani@sheffield.ac.uk)

**Abstract.** The second-order discontinuous Galerkin (DG2) solver of the shallow water equations in LISFLOOD-FP 8.0 is well-suited for predicting small-scale transients that emerge in rapid, multiscale floods caused by impact events like tsunamis. However, this DG2 solver can only be used for simulations on a uniform grid where it may yield inefficient runtimes even when using its graphics processing unit (GPU) parallelised version (GPU-DG2). To maximise runtime reduction, the new LISFLOOD-FP 8.2 version integrates GPU parallelised dynamic (in time) grid resolution adaptivity of

multiwavelets (MW) with the DG2 solver (GPU-MWDG2). The GPU-MWDG2 solver requires selecting a maximum refinement level, $L$, based on size and resolution of the Digital Elevation Model (DEM) and an error threshold, $\varepsilon \leq 10^{-3}$, to preserve similar accuracy as a GPU-DG2 simulation on a uniform grid. The accuracy and efficiency of dynamic GPU-MWDG2 adaptivity is assessed for four tsunami-induced flooding test cases involving increasingly complex tsunamis: from single-wave impact events to wave trains. At $\varepsilon = 10^{-3}$, the GPU-MWDG2 simulations yield predictions similar to the GPU-

DG2 simulations, but using $\varepsilon = 10^{-4}$ can improve the accuracy in velocity-related predictions. In terms of efficiency, the GPU-MWDG2 simulations show progressively larger speedups over the GPU-DG2 simulations from $L \geq 10$, which become significant ($\geq$ 3.3- and 4.5-fold at $\varepsilon = 10^{-4}$ and $10^{-3}$, respectively) for simulating a single-wave impact event. The LISFLOOD-FP 8.2 code is open source, DOI: 10.5281/zenodo.4073010, as well as the simulation data and the input files and scripts to reproduce them, DOI: 10.5281/zenodo.13909072, with additional documentation at

https://www.seamlesswave.com/Adaptive (last accessed: 9 October 2024).



## 1 Introduction

LISFLOOD-FP is a raster-based hydrodynamic modelling framework that has been used to support various geoscientific modelling applications (e.g. Hajihassanpour et al., 2023; Hunter et al., 2005; Nandi & Reddy, 2022; Zeng et al., 2022; Ziliani et al., 2020). LISFLOOD-FP has a suite of numerical solvers of the two-dimensional shallow water equations, including a

diffusive wave solver (Hunter et al., 2005), a local inertial solver (Bates et al., 2010), a first-order finite volume solver, and a second-order discontinuous Galerkin (DG2) solver (Shaw et al., 2021). The DG2 solver is the most complex numerically, requiring three times more degrees of freedom per computed variable and at least twelve times more computations per cell compared to any of the other solvers in LISFLOOD-FP (Ayog et al., 2021; Kesserwani et al., 2018; Shaw et al., 2021). Even when parallelised on a graphics processing unit (GPU), the GPU parallelised DG2 solver (GPU-DG2) may still exhibit

prohibitively long runtimes when used to run real-world flood simulations on Digital Elevation Models (DEMs) with raster grid sizes beyond the kilometre scale and/or at grid resolutions near or below the metre scale (Kesserwani & Sharifian, 2023; Shaw et al., 2021).

Although DG2 simulations are capable of accurately reproducing slow to gradual flooding flows, even at very coarse DEM resolutions (Ayog et al., 2021; Kesserwani, 2013; Kesserwani & Wang, 2014; Shaw et al., 2021), they

primarily excel at capturing small-scale transients that occur over a wide range of spatial and temporal scales (Kesserwani et al., 2023; Kesserwani & Sharifian, 2023; Sharifian et al., 2018; Sun et al., 2023). Such transients are typical of rapid flooding flows triggered and driven by multiscale impact event(s) like tsunami(s), which can include zones of flow recirculation past unsubmerged island(s). Hence, DG2 simulations are well-suited for obtaining detailed modelling of rapid, multiscale flooding flows, such as in tsunami-induced flooding. Within this scope for the modelling, dynamic (in time) mesh adaptivity

has often been deployed with finite volume based tsunami inundation simulators to reduce simulation runtimes (e.g. Lee, 2016; Popinet, 2012). This paper reports the integration of dynamic grid resolution adaptivity with the GPU-DG2 solver in LISFLOOD-FP to reduce the runtimes of rapid multiscale flow simulations, which are exemplified by tsunami-induced flooding events.

Unlike static grid resolution adaptivity, which was integrated into LISFLOOD-FP 8.1 (Sharifian et al., 2023), this

LISFLOOD-FP 8.2 version performs dynamic grid resolution adaptivity every simulation timestep to achieve as much local grid resolution coarsening as possible for grid cells covering regions of smooth flow and DEM features, thereby reducing the computational effort and runtime of a simulation by reducing the number of computational cells in the grid. The LISFLOOD-FP 8.2 version is unique in providing a single, mathematically sound hydrodynamic modelling framework that integrates dynamic grid resolution adaptivity on the GPU for achieving raster-grid DG2 simulations that can preserve a

similar level of predictive accuracy and robustness as alternative GPU-DG2 simulations on a uniform grid. In fact, existing hydrodynamic modelling frameworks for tsunami simulation that have integrated dynamic grid resolution adaptivity with GPU parallelisation were mostly based on finite volume simulators (Berger et al., 2011; de la Asunción & Castro, 2017; Ferreira & Bader, 2017; Kevlahan & Lemarié, 2022; LeVeque et al., 2011; Liang et al., 2015; J. Park et al., 2019; Popinet,



2011, 2012; Popinet & Rickard, 2007). Comparatively, there are fewer simulators based on DG methods, mostly in the context of tsunami inundation simulation and considering triangular or curvilinear meshes, with sparse focusses: either on integrating central processing unit (CPU) parallelisation of extrinsic forms of dynamic adaptivity, or on parallelising (non-adaptive) DG simulators on the GPU; while, in any of the focusses, addressing the robustness treatments for the integration of wet-dry fronts and/or the source terms (Blaise et al., 2013; Blaise & St-Cyr, 2012; Bonev et al., 2018; Castro et al., 2016; Hajihassanpour et al., 2019).

To mention just a few, Blaise & St Cyr (2012) and Blaise et al. (2013) integrated CPU parallelisation with dynamic adaptivity for curvilinear meshes, calling for better forms of adaptivity with better robustness treatments to achieve reliable DG based tsunami inundation simulations. Rannabauer et al. (2018) addressed wet-dry front treatments with a DG based simulator of tsunami inundation that integrated CPU parallelised dynamic adaptivity on triangular meshes; further, the authors identified the benefit of their DG based simulator in comparison with a finite volume simulator. To track tsunami propagation on the sphere, Bonev et al. (2018) and Hajihassanpour et al. (2019) developed DG based simulators with dynamic adaptivity for curvilinear meshes, highlighting the need to further exploit GPU parallelisation to achieve practical runtimes. For tsunami inundation simulations, Castro et al. (2016) found that non-adaptive DG simulator on triangular meshes yield 23-fold faster runtimes when parallelised on the GPU as compared to when parallelised on the CPU with 24 threads. Yet, to the best of the writers' knowledge, there is no existing DG based hydrodynamic modelling framework that combines raster grid-based dynamic resolution adaptivity with GPU parallelisation packed within a mathematically sound framework that intrinsically preserves the robustness and predictive accuracy of the DG solver on the uniform grid. In this paper, such a GPU parallelised adaptive hydrodynamic modelling framework is optimised and newly integrated into LISFLOOD-FP 8.2; the framework combines dynamic grid resolution adaptivity of multiwavelets (MW) with the DG2 solver formulation – this combination is, hereafter, referred to as dynamic GPU-MWDG2 adaptivity or the GPU-MWDG2 solver.

Dynamic MWDG2 adaptivity automates local grid resolution coarsening on a raster-based adaptive grid via the multiresolution analysis (MRA) of MW applied to scaled DG2 modelled data – considering both (time-varying) flow solutions and (time-invariant) DEM representations (Kesserwani et al., 2019; Kesserwani & Sharifian, 2020; Sharifian et al., 2019, 2023). As the scaling, analysis, and reconstruction of DG2 modelled data are all inherent to the MRA procedure, the existing robustness treatments incorporated in the reference GPU-DG2 solver are readily preserved irrespective of the variability in the resolution scales. Another benefit of the MRA procedure is the reliance on a single criterion, an error threshold $\varepsilon$, to sensibly control the amount of local grid resolution coarsening. For $\varepsilon \leq 10^{-3}$, dynamic MWDG2 adaptivity was shown to preserve similarly accurate simulations as the reference DG2 solver run on the uniform grid (Caviedes-Voullième et al., 2020; Caviedes-Voullième & Kesserwani, 2015; Gerhard et al., 2015; Kesserwani et al., 2015); and Kesserwani & Sharifian (2020) formulated CPU-based MWDG2 solvers that further preserve the robustness of a reference DG2 solver designed for realistic, two-dimensional hydrodynamic modelling (Kesserwani et al., 2018).





Chowdhury et al. (2023) devised an efficient GPU parallelisation of wavelet adaptivity for finite volume hydrodynamic simulations. Their results show that the speedup afforded by wavelet adaptivity scales up with the maximum refinement level, $L$ – selected from the size and resolution of the raster-formatted DEM file – and starts offering positive speedups over uniform-grid finite volume GPU simulations starting from $L \geq 9$. Kesserwani & Sharifian (2023) extended the GPU parallelisation of wavelet adaptivity to produce a GPU-MWDG2 solver and analysed its efficiency using $\varepsilon = 10^{-3}$ in simulating realistic slow-to-rapid flooding flow scenarios that involved $L \geq 10$. Their findings revealed that the GPU-MWDG2 solver is three times faster than the GPU-DG2 solver when simulating a rapid flood scenario driven by an impact event, requiring $L = 11$, as long as the dynamic GPU-MWDG2 adaptivity does not use more than 85 % the number of cells on the uniform grid of the GPU-DG2 simulation. These findings motivate for a dedicated study about the potential speedup of GPU-MWDG2 simulations over GPU-DG2 simulations in the context of rapid multiscale flooding scenarios involving $L \geq 9$ and $\varepsilon \leq 10^{-3}$.

Next, in Sect. 2, the GPU-MWDG2 solver in LISFLOOD-FP 8.2 is described with a focus on its use for running GPU-MWDG2 simulations from raster-formatted DEM and initial flow setup files (Sect. 2.1), its associated upper memory limits (Sect. 2.2), and its efficiency analysis using several proposed metrics obtained from postprocessing simulation outputs (Sect. 2.3). In Sect. 3, the efficiency of the GPU-MWDG2 solver using $\varepsilon = 10^{-3}$ and $10^{-4}$ is assessed with reference to the GPU-DG2 solver by considering four test cases of tsunami-induced flooding with increasingly complex tsunamis, from single-wave impact events to wave trains. Sect. 4 draws conclusions and recommendations as to when the GPU-MWDG2 solver can best lead to considerable speedups over the GPU-DG2 solver. The LISFLOOD-FP 8.2 code is open-source under the GPL v3.0 licence (LISFLOOD-FP developers, 2024) in addition to the simulation results and the input files and scripts to reproduce them (Chowdhury & Kesserwani, 2024), with further guidance at https://www.seamlesswave.com/Adaptive (last accessed: 9 October 2024).

## 2 LISFLOOD-FP 8.2

LISFLOOD-FP 8.2 includes the new capability of running simulations over a non-uniform grid using dynamic GPU-MWDG2 adaptivity. The GPU-MWDG2 solver can be used as an alternative to the uniform-grid GPU-DG2 solver (Shaw et al., 2021) to potentially reduce simulation runtimes. Unlike with LISFLOOD-FP 8.1, where the MRA procedure of MW is only applied once at the beginning of the simulation to generate a static non-uniform grid whose grid resolution is locally coarsened as much as permitted by features of the DEM that are time-invariant (Sharifian et al., 2023), the GPU-MWDG2 solver deploys the MRA procedure every simulation timestep, denoted by $\Delta t$, to also automate grid resolution coarsening based on the features of the time-varying flow solution.

The algorithmic description of the GPU-MWDG2 solver has been reported in previous papers (Kesserwani & Sharifian, 2020; 2023), which its dynamic adaptivity has been further optimised to improve memory coalescing and occupancy in the GPU kernels (NVIDIA, 2023). Therefore, the GPU-MWDG2 solver is only briefly overviewed here (see





Appendix A) with a focus on its operational workflow, shown in Figure 1. Here, the presentation is focussed on describing

the features incorporated into LISFLOOD-FP 8.2 for running the GPU-MWDG2 solver (Sect. 2.1), on identifying the upper

limits of its GPU memory consumption in relation to the specification of the GPU card (Sect. 2.2), and on proposing metrics

for detailed analysis of the efficiency of its dynamic adaptivity from output datasets (Sect. 2.3).

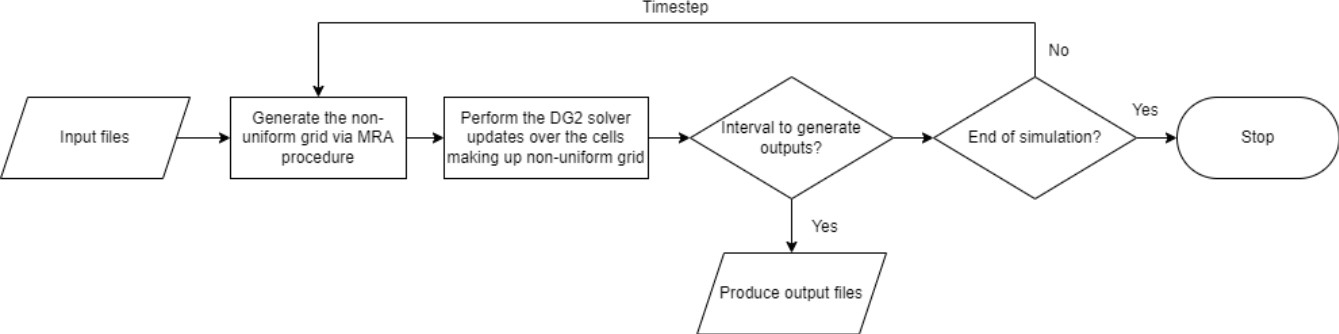

**Figure 1**: The main operations involved in the GPU-MWDG2 solver (further detailed in Appendix A).

## 2.1 The GPU-MWDG2 solver

Running a simulation of a test case using any solver in LISFLOOD-FP requires setting up several test case-specific input

files[1], and the same is required for the GPU-MWDG2 solver. An important input file is the "parameter" file with the

extension .par, which is a text file specifying various solver and simulation parameters[2]. In the remainder of this paper, the

usability of the GPU-MWDG2 solver will be described for the "Monai valley" test case (explored in Sect. 3.1) – without loss

of generality. Step-by-step instructions on how to use the GPU-MWDG2 solver to run a simulation of the "Monai valley"

test case have been provided in Appendix B.

```
 1  cuda
 2  mwdg2
 3  epsilon      0.001
 4  max_ref_lvl  9
 5  wall_height  0.5
 6  refine_wall
 7  ref_thickness 16
 8  initial_tstep 1
 9  cumulative
10  raster_out
11  fpfric       0.01
12  sim_time     22.5
13  massint      0.1
14  saveint      22.5
15  DEMfile      monai.dem
16  startfile    monai.start
17  bcifile      monai.bci
18  bdyfile      monai.bdy
19  stagefile    monai.stage
```

**Figure 2**: Listing of parameters in the .par file needed to run a GPU-MWDG2 simulation for the "Monai Valley" test case

(Sect. 3.1), with the GPU-MWDG2 specific items highlighted in bold.

---

[1] https://www.seamlesswave.com/Merewether1; https://www.seamlesswave.com/Adaptive

[2] https://www.seamlesswave.com/Merewether1-1.html




The parameters[2] or keywords that should be typed in the `.par` file for running a simulation of the Monai Valley test case are shown in Figure 2, including seven keywords related to running the GPU-MWDG2 solver highlighted in bold. The `cuda` keyword should be typed to access the GPU parallelised models in LISFLOOD-FP, e.g. the GPU-DG2 solver or the GPU-MWDG2 solver. The `mwdg2` keyword should be typed to select the GPU-MWDG2 solver[3]. The `epsilon` keyword

followed by a numerical value 0.001 specifies the error threshold $\varepsilon = 10^{-3}$. The `max_ref_lvl` keyword followed by an integer value specifies the maximum refinement level $L$, specified according to the DEM size and resolution as explained next.

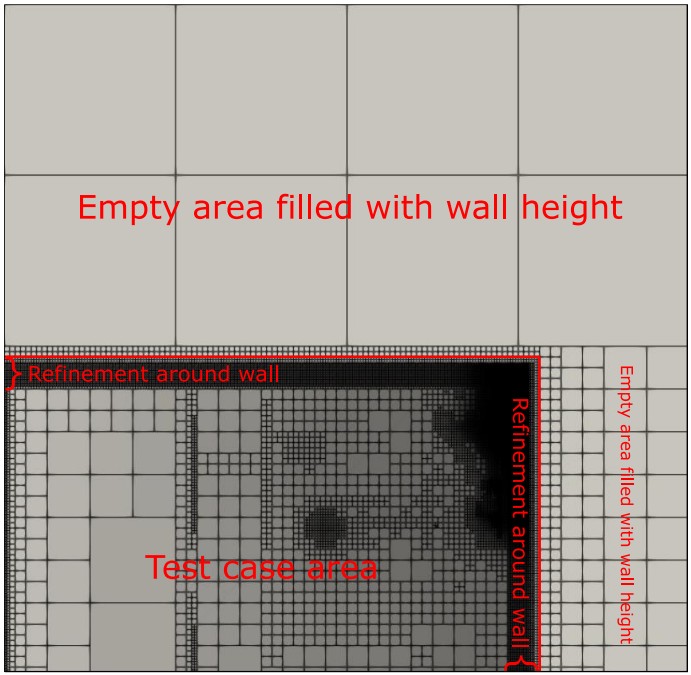

**Figure 3**: Initial non-uniform grid generated by the GPU-MWDG2 solver, via the MRA procedure, based on the static
features the bathymetry for the "Monai Valley" test case (Sect. 3.1).

The GPU-MWDG2 solver starts a simulation on a *square uniform grid* made up of $2^L \times 2^L$ cells, which is the finest-resolution grid accessible to the GPU-MWDG2 solver (Appendix A). Practically, the DEM often involves a (rectangular) grid with $M$ rows and $N$ columns, for which the GPU-MWDG2 solver should still generate a starting *square uniform grid*

with $2^L \times 2^L$ cells. The most optimal choice would be by selecting the smallest value of $L$ such that $2^L \geq \max(N, M)$. For example, in the Monai valley test case, $N = 784$ and $M = 486$, and $L$ should be the smallest integer such that $2^L \geq \max(784, 486)$, leading to the choice $L = 10$. Figure 3 shows the initial non-uniform grid generated by the GPU-MWDG2 solver. Since the GPU-MWDG2 solver starts from a *square uniform grid* inclusive of the DEM dimensions, two areas

---

[3] The CPU version of the MWDG2 solver was not integrated on LISFLOOD-FP due to its uncompetitive runtimes.





emerge in the non-uniform grid: the actual test case area, which includes the DEM data and the initial flow conditions; and,

empty areas where no DEM data are available and where no flow should occur.

In the actual test case area, GPU-MWDG2 initialises the data in the cells by using the values specified in `.dem` and `.start` files in raster grid format (see Appendix B). Meanwhile, in the empty areas, it initialises the flow data to zero and assigns bathymetry data the numerical value that follows the `wall_height` keyword. This numerical value must be sufficiently high such that a wall is generated between the test case area and the empty areas (see Figure 3) that prevents any

water from leaving the test case area (e.g. by choosing a numerical value that is higher than the largest water surface elevation). For the Monai valley test case, the `wall_height` keyword is specified to 0.5 m to generate a wall that is high enough to prevent any water from leaving the test case area. The `refine_wall` keyword and the `ref_thickness` keyword, followed by an integer for the latter, typically between 16 and 64, should also be typed in the parameter file to prevent GPU-MWDG2 from excessively coarsening the non-uniform grid around the walls (labelled with the curly braces in

Figure 3). For the Monai valley test case, the `refine_wall` keyword is specified to trigger refinement around the wall, and `ref_thickness` is specified as 16 to trigger 16 cells at the highest refinement level between the wall and the test case area.

The remaining keywords in Figure 2 are standard for running simulations using LISFLOOD-FP and were described previously[2]. Note that running GPU-MWDG2 on LISFLOOD-FP 8.2 only requires the user to provide the `.dem` file and `.start` files, unlike the DG2 solvers in LISFLOOD-FP 8.0 (Shaw et al., 2021) and the static non-uniform grid generator in

LISFLOOD-FP 8.1 (Sharifian et al., 2023), which require providing `.dem1x`, `.dem1y`, `.start1x` and `.start1y` raster files to initialise the slope coefficients of the DG2 solver. These coefficients are automatically initialised by the GPU-MWDG2 solver in LISFLOOD-FP 8.2.

Compared to a GPU-DG2 simulation, a GPU-MWDG2 simulation consumes much more memory. As shown next in Sect. 2.2, the large memory costs arise from the need to store the objects involved in the GPU-MWDG2 algorithm.

Practically, the largest allowable choice of $L$, or largest *square uniform grid*, is restricted by the memory capacity of the GPU card on which the GPU-MWDG2 simulation is performed.

## 2.2 GPU memory cost analysis and limits

The scope for running a GPU-MWDG2 simulation depends on the availability of a GPU card that can fit the memory costs for the specified choice of $L$. The left panel in Figure 4 shows the breakdown percentage of the memory consumed by the

objects involved in GPU-MWDG2 simulations, i.e. the GPU-MWDG2 non-uniform grid, the explicit neighbours of each cell in the grid, and the hierarchy of grids involved in the dynamic GPU-MWDG2 adaptivity process (overviewed in Appendix A). It can be seen that 15% of the memory is allocated for arrays storing the hierarchy of uniform grids, and 6% is allocated for other miscellaneous purposes. Remarkably however, nearly 80% of the overall GPU memory costs are due to allocating arrays for the non-uniform grid and its neighbours: 22% from storing the array representing the non-uniform grid, while

another 57% from explicitly storing the four neighbours of each cell in the non-uniform grid. This is because the GPU-





MWDG2 solver is coded to allocate GPU memory for the worst-case scenario where there is no grid coarsening at all, thereby negating the need for memory reallocation after any coarsening to maximise the efficiency of dynamic GPU-MWDG2 adaptivity, since memory allocation is a relatively slow operation.

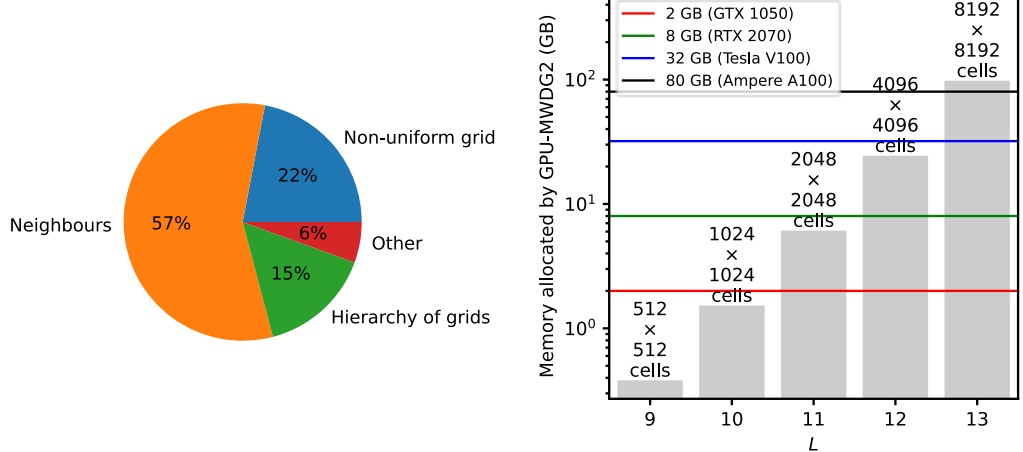

**Figure 4:** GPU memory consumed by dynamic GPU-MWDG2 adaptivity. Left panel shows the percentage breakdown of the GPU memory consumed by the different objects involved in the GPU-MWDG2 solver. Right panel shows the amount of GPU memory allocated against the maximum refinement level $L$; The numbers on top of the bars show the number of cells for a given value of $L$. The horizontal lines indicate the memory limits of four GPU cards.

The right panel in Figure 4 displays the GPU memory allocated by the GPU-MWDG2 simulations for different $L$ leading to $2^L \times 2^L$ cells on the *square uniform grid*. The coloured lines represent the memory limits of four different GPU cards. In this figure, the memory limits are considered for $L \geq 9$, i.e. for the case where wavelet-based adaptive simulations were shown to start offering speedups over the uniform-grid simulations (Chowdhury et al., 2023; Kesserwani & Sharifian, 2023). As can be seen, dynamic GPU-MWDG2 adaptivity can only allocate GPU memory below the upper memory limit of the GPU card under consideration, leading to a restriction on the value of $L$ that can be employed. For instance, a GTX 1050 card with a memory capacity of 2 GB can only accommodate GPU-MWDG2 simulations up to $L = 10$, i.e. starting from a *square uniform grid* made of $1024 \times 1024$ cells; this is because any value of $L > 10$ will lead to exceeding this GPU card's memory limit. Generally, the larger the value of $L$, the larger the $2^L \times 2^L$ cells on the *square uniform grid*, thus the larger the memory requirement for the GPU card. At the time this study was conducted, GPU-MWDG2 simulations involving $L \geq 13$, i.e. starting from a *square uniform grid* from $8192 \times 8192$ cells, were not feasible because accommodating such values of $L$ needed >80 GB of GPU memory, which was higher than the memory limit of the latest commercially available GPU card (i.e. the A100 GPU card, with 80 GB of memory).




### 2.3 Metrics for analysing GPU-MWDG2's runtime efficiency

Assessing the potential speedup that could be afforded by GPU-MWDG2 adaptivity over a GPU-DG2 simulation is essential

for a user. As noted in other works that have explored wavelet adaptivity, the computational effort and speedup of a GPU-MWDG2 simulation should ideally be correlated exactly with the number of cells in the GPU-MWDG2 non-uniform grid (since the number of cells dictates the number of DG2 solver updates to be performed). However, in practice, this rarely occurs as the ideal speedup is diminished by the additional computational effort spent by GPU-MWDG2 to generate the non-uniform grid every timestep via the MRA process (Kesserwani et al. 2019; Kesserwani and Sharifian, 2020; Kesserwani and

Sharifian, 2023; Chowdhury et al. 2023). Thus, to thoroughly assess the potential speedup of a GPU-MWDG2 simulation, the user must consider the interdependent effects of the number of cells in the non-uniform grid, the computational effort of performing the DG2 solver updates, and the computational effort of performing the MRA process.

To this end, starting from LISFLOOD-FP 8.2, the user can include the "cumulative" keyword in the parameter file to produce a ".cumu" file that contains the time histories of several quantities for analysing the speedup achieved by

GPU-MWDG2 adaptivity [i.e. the number of cells in the non-uniform grid, the computational effort of performing the DG2 solver updates per timestep, the timestep size, the timestep count, amongst other items with the full list of items detailed in any of the the data and script files in Chowdhury and Kesserwani (2024)]. In this paper, the time histories of these quantities are postprocessed into several time-dependent metrics for analysing the speedups of GPU-MWDG2 simulations compared to GPU-DG2 simulations (Sect. 3). The metrics are described in Table 1, and their use for analysing the speedup of a GPU-

MWDG2 simulation is explained next by way of an example.

**Table 1**: Time-dependent metrics for evaluating the potential speedup of a GPU-MWDG2 simulation over a GPU-DG2 simulation.

| Metric | Description |
|---|---|
| $N_{cells}(t)$ | Number of cells in GPU-MWDG2's non-uniform grid compared to GPU-DG2's grid (as a percentage) against simulation time. |
| $R_{DG2}(t)$ | Computational effort spent by GPU-MWDG2 to perform the DG2 solver updates at a given timestep (relative to GPU-DG2 as a percentage) against simulation time. |
| $R_{MRA}(t)$ | Computational effort spent by GPU-MWDG2 to perform the MRA process and generate the non-uniform grid at a given timestep (relative to GPU-DG2 as a percentage) against simulation time. |
| $S_{inst}(t)$ | Instantaneous speedup achieved by GPU-MWDG2 over GPU-DG2 at a given timestep against simulation time. |
| $N_{\Delta t}(t)$ | Number of timesteps taken by GPU-MWDG2 to reach a given simulation time. |
| $C_{DG2}(t)$ | Cumulative computational effort spent by GPU-MWDG2 to perform the DG2 solver updates (quantified in units of wall clock time) up to a given simulation time. |





| $C_{MRA}(t)$ | Cumulative computational effort spent by GPU-MWDG2 to perform the MRA process (quantified in units of wall clock time) up to a given simulation time. |
|---|---|
| $C_{tot}(t)$ | Total cumulative computational effort spent by GPU-MWDG2 to complete a simulation (quantified in units of wall clock time) up to a given simulation time. |
| $S_{acc}(t)$ | Accumulated speedup of GPU-MWDG2 over GPU-DG2 up to a given simulation time. |

In a GPU-MWDG2 simulation of an impact event, the computational effort per timestep changes depending on the change in the number of cells in the GPU-MWDG2 non-uniform grid. The number of cells changes over time because finer cells are generated by GPU-MWDG2 adaptivity to track the flow features produced by the impact event as it enters and travels through the bathymetric area. Using the same Monai Valley example (Sect 3.1), the left panel of Figure 5 shows the initial non-uniform grid generated by GPU-MWDG2 at the start of the simulation, while the right panel shows an

intermediate non-uniform grid generated by GPU-MWDG2 after the simulation has progressed by 17 s, i.e. after an impact event, here a tsunami, has entered and propagated through the bathymetric area. At the start of the simulation, the initial non-uniform grid is coarsened as much as allowed, based only on the static features of the bathymetric area and initial flow conditions, leading to a minimal number of cells in the grid, which is quantified by $N_{cell}$. The number of cells determines the number of DG2 solver updates to be performed at a given timestep, leading to a corresponding computational effort per

timestep, which is quantified by $R_{DG2}$. There is also the computational effort of performing the MRA process at a given timestep, which is quantified by $R_{MRA}$. Based on the combined computational effort of performing both the MRA process and the DG2 solver updates at a given timestep, the *instantaneous* speedup in completing one timestep of a GPU-MWDG2 simulation can be computed (relative to the GPU-DG2 simulation), which is quantified by $S_{inst}$. In Figure 5, after the simulation has progressed by 17 s, the number of cells in the non-uniform grid has increased due to using finer cells to track

the tsunami's wavefronts and wave diffractions, which leads to a higher value of $N_{cell}$ and $R_{DG2}$ (and possibly also to a higher value of $R_{MRA}$, as a higher number of cells in the non-uniform grid means more cells must be processed during the MRA process); thus, $S_{inst}$ is expected to drop. Generally, the higher the complexity of the impact event, the higher the number of cells in the GPU-MWDG2 non-uniform grid, and the lower the potential speedup.





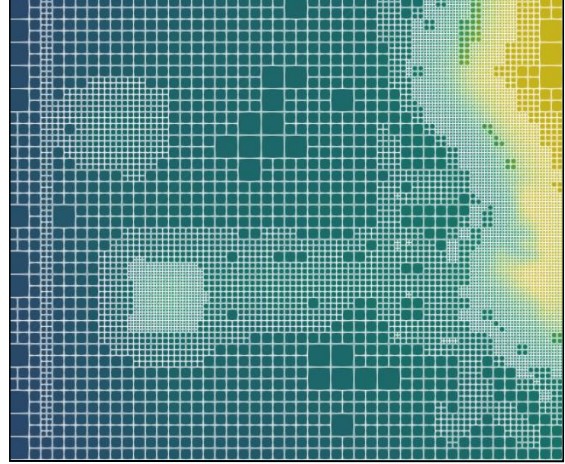 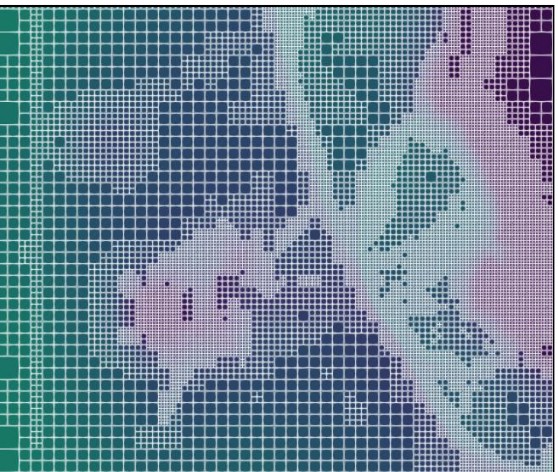

**Figure 5**: GPU-MWDG2 non-uniform grids generated for the "Monai Valley" test case (Sect. 3.1). Left panel shows the grid at the start of the simulation whereas the right panel shows the grid after the simulation has progressed by 17 s, which tracks flow dynamics.

The metrics $N_{cell}$, $R_{DG2}$, $R_{MRA}$ and $S_{inst}$ quantify the computational effort and speedup of a GPU-MWDG2 simulation per timestep compared to a GPU-DG2 simulation. However, the overall or *cumulative* computational effort and speedup of a GPU-MWDG2 simulation depends on having accumulated the computational effort and speedup per timestep from *all* the timesteps taken by GPU-MWDG2 to reach a given simulation time. The higher the number of timesteps taken by GPU-MWDG2 to reach a given simulation time (quantified by $N_{\Delta t}$), the higher the cumulative computational effort spent by GPU-MWDG2 to reach that simulation time (quantified by $C_{tot}$). The $C_{tot}$ metric is computed by summing the cumulative computational effort spent by GPU-MWDG2 to perform the DG2 solver updates and the MRA process, which is quantified by $C_{DG2}$ and $C_{MRA}$, respectively. Using the cumulative metrics, the overall speedup accumulated by a GPU-MWDG2 simulation can be computed, which is quantified by $S_{acc}$. The metrics in Table 1 are used in the next section (Sect. 3) to assess the speedup afforded by GPU-MWDG2 adaptivity.

## 3 Evaluation of GPU-MWDG2 adaptivity

The efficiency of the GPU-MWDG2 solver is evaluated against the GPU-DG2 solver (which is always run on a uniform grid at the finest resolution used by the GPU-MWDG2 solver) by running and comparing GPU-MWDG2 simulations against GPU-DG2 simulations using the time-dependent metrics proposed in Table 1 (Sect. 2.3). For completeness, the accuracy of GPU-MWDG2 simulations is also evaluated, namely by quantifying the difference between the predictions of a GPU-MWDG2 simulation and a GPU-DG2 simulation using the root mean squared error (RMSE) and the correlation coefficient ($r$), which are given by:





$$RMSE = \sqrt{\frac{\sum_i^{N_s} (P_{i,MWDG2} - P_{i,DG2})^2}{N_s}} \qquad (1)$$

$$r = \frac{\sum_i^{N_s} (P_{i,MWDG2} - \overline{P_{MWDG2}})(P_{i,DG2} \, \overline{P_{DG2}})}{\sqrt{\left[\frac{1}{N_s}\sum_i^{N_s}(P_{i,MWDG2} - \overline{P_{MWDG2}})^2\right]\left[\frac{1}{N_s}\sum_i^{N_s}(P_{i,DG2} - \overline{P_{DG2}})^2\right]}} \qquad (2)$$

where, $N_s$ denotes the number of sampling points, $P_{i,MWDG2}$ and $P_{i,DG2}$ are the $i$th points where GPU-MWDG2 and GPU-DG2 predictions are spatially and/or temporally sampled, respectively, and $\overline{P_{MWDG2}}$ and $\overline{P_{DG2}}$ are their mean predictions across all

280 sampling points. The RMSE measures the closeness between the predictions, whereas the $r$ value measures the correlation (similarity) among these same predictions. The nearer to 0 the RMSE, the closer the predictions, and the nearer to 1 the $r$ value, the higher their similarity. The GPU-MWDG2 simulations are run with $\varepsilon = 10^{-4}$ and $10^{-3}$, which are the values for which the GPU-MWDG2 solver preserves the predictive accuracy of the GPU-DG2 solver while achieving a fair level of efficiency (Kesserwani et al., 2019; Kesserwani & Sharifian, 2020; 2023; Sharifian et al., 2019; 2023).

**Table 2**: Characteristics of the four selected test cases listed in order of tsunami complexity and including the DEM size and resolution dictating the choice of $L$, the Manning coefficient, $n_M$, and the simulation output time, $t_{end}$.

| Test case | DEM size | Tsunami complexity | $L$ | $t_{end}$ | $n_M$ |
|---|---|---|---|---|---|
| "Monai Valley" (Sect. 3.1) | 784 rows × 486 columns (0.007 m resolution) | Single-wave event with a smooth wave peak during 10 and 15 s, followed by a trough, during 15 and 20 s. | 10 | 22.5 s | 0.01 |
| "Seaside Oregon" (Sect. 3.2) | 2181 rows × 1091 columns (0.02 m resolution) | Single-wave event with a long wave, without a trough, that occurs by 10 s, travelling towards the coast, from 15 s, to then hit a complex urban town. | 12 | 40 s | 0.025 |
| "Tauranga Harbour" (Sect. 3.3) | 4096 rows × 2196 columns (10 m resolution) | Wave train event of three low-frequency waves with troughs (two including noise) propagating over a long duration of 40 hr. | 12 | 40 hr | 0.025 |
| "Hilo Harbour" (Sect. 3.4) | 702 rows × 692 columns (10 m resolution) | Wave train event with many high-frequency waves with troughs propagating over a long duration of 6 hr. | 10 | 6 hr | 0.025 |

As the potential speedup afforded by GPU-MWDG2 adaptivity is hypothesised to depend on the impact event

complexity and the DEM size (dictating the choice of the $L$), the speedup evaluation is performed using four realistic tsunami-induced flooding test cases that each feature a unique combination of impact event complexity (either a simple single-wave tsunami or a complex wave train tsunami) and DEM size (requiring either $L = 10$ or 12). Table 2 includes the test case-specific DEM sizes and resolutions that dictate the choice of $L \geq 10$ and the physical set-up parameters, i.e. the

 

Manning coefficient, $n_\text{M}$, and the simulation end time, $t_\text{end}$. The GPU-MWDG2 and GPU-DG2 simulations were run on the
Stanage high performance computing cluster of the University of Sheffield to access the A100 GPU card with 80 GB of
memory, necessary for accommodating the memory costs of GPU-MWDG2 simulations requiring an $L$ value as high as 12
(Sect. 2.2).

### 3.1 Monai Valley

This test case was used to validate many hydrodynamic solvers (Caviedes-Voullième et al., 2020; Kesserwani & Liang,
2012; Kesserwani & Sharifian, 2020; Matsuyama & Tanaka, 2001). It involves a 1:400 scaled replica of the 1993 tsunami
that flooded Okushiri Island after a wave runup of 30 m at the tip of a very narrow gulley in a small cove at Monai Valley
(Liu et al., 2008). The scaled DEM has $784 \times 486$ cells for which its associated initial *square uniform grid* is generated with
$L = 10$.

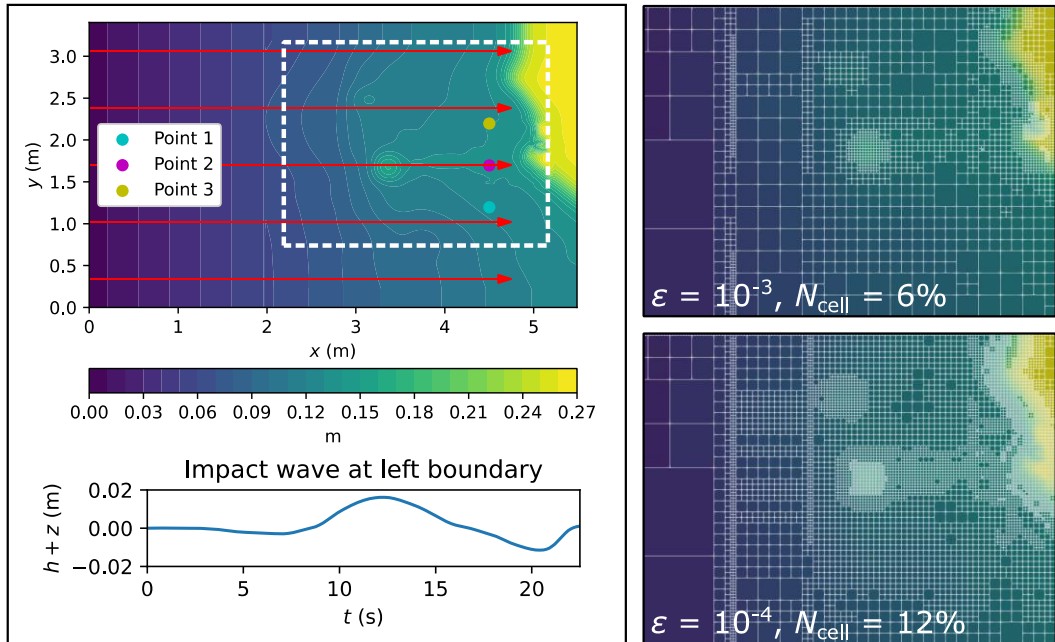

**Figure 6:** Monai Valley. Top-down view of bathymetry (top left panel), where the red arrows indicate the direction and
distance travelled by the tsunami; time history of the tsunami entering from the left boundary (bottom left panel); initial
GPU-MWDG2 non-uniform grids (right panels) covering the portion of the bathymetric area framed by the white box (top
left panel).

In Figure 6, a top-down view of the bathymetric area is shown (top left panel), which has a small island in the
middle and a coastal shoreline to the right, including Okushiri Island and Monai Valley. The coastal shoreline gets flooded
by a tsunami that initially enters the bathymetric area from the left boundary and then travels to the right by 4.5 m (indicated
by the red arrows), interacting with the small island as it travels through the bathymetric area. This tsunami is simulated for




22.5 s, during which it travels through the bathymetric area in three stages of flow over time: the entry stage (0 to 7 s), the

travelling stage (7 to 17 s) and the flooding stage (17 to 22 s). During the entry stage, the tsunami does not enter the

bathymetric area, as seen in the hydrograph of the tsunami's water surface elevation (bottom left panel of Figure 6). During

the travelling stage, the tsunami enters from the left boundary and travels right towards the coastal shoreline. Lastly, during

the flooding stage, the tsunami floods the coastal shoreline, due to which many flow dynamics such as wave reflections and

diffractions are produced that must be tracked using finer cells, thus increasing the number of cells in the GPU-MWDG2

non-uniform grid. In the right panels of Figure 6, the initial GPU-MWDG2 grids at $\varepsilon = 10^{-3}$ and $10^{-4}$ are depicted for the

portion of the bathymetric area framed by the white box (top left panel of Figure 6). With $\varepsilon = 10^{-3}$ and $10^{-4}$, the initial GPU-

MWDG2 grid has 6% and 12% of the number of cells as in the GPU-DG2 uniform grid, respectively.

**Figure 7:** Monai Valley. Metrics of Table 1 applied to the GPU-MWDG2 and GPU-DG2 simulations. Also shown is a time
history of $\Delta t$ (centre panel).





In Figure 7, an analysis of the runtimes of the GPU-DG2 and GPU-MWDG2 simulations using the time-dependent metrics of Table 1 is shown; a time history of $\Delta t$ is also included. The dashed lines in the top left panel indicate the initial value of $N_{cell}$, i.e. $N_{cell}$ of the initial GPU-MWDG2 non-uniform grids. Up to 15 s, i.e. before the flooding stage of flow

begins, the time history of $N_{cell}$ remains flat, meaning that the number of cells in the GPU-MWDG2 non-uniform grid does not change over time. Once the flooding stage begins however, $N_{cell}$ increases slightly, particularly at $\varepsilon = 10^{-4}$. With an increased number of cells in the non-uniform grid, the computational effort of performing the DG2 solver updates per timestep should increase, which is confirmed by the time history of $R_{DG2}$, which is flat before the flooding stage of flow, but thereafter increases, particularly at $\varepsilon = 10^{-4}$. Conversely, unlike $R_{DG2}$, the time history of $R_{MRA}$ is similar for both values of $\varepsilon$,

and stays flat for most of the simulation except for an initial decrease at the start, meaning that the computational effort of performing the MRA process per timestep is similar for both values of $\varepsilon$ and remains fixed throughout the simulation. Thus, the drop in the speedup of completing a single timestep of the GPU-MWDG2 simulation compared to the GPU-DG2 simulation is mostly due to the increase in $R_{DG2}$ at $\varepsilon = 10^{-4}$, with $S_{inst}$ dropping from 2.0 to 1.8 (which otherwise stays flat at 2.7 for $\varepsilon = 10^{-3}$).

Besides analysing the computational effort and speedups of the GPU-MWDG2 simulations per timestep, there is also the question of analysing the cumulative computational effort of running the simulations, which depends on the timestep size ($\Delta t$) and the number of timesteps taken to reach a given simulation time ($N_{\Delta t}$). In this test case, the time histories of $\Delta t$ of the GPU-DG2 simulation and the GPU-MWDG2 simulation using $\varepsilon = 10^{-4}$ are very similar, but with $\varepsilon = 10^{-3}$, $\Delta t$ drops at 7 s, i.e. as soon as the travelling stage of flow begins. This slight drop in $\Delta t$ is likely dictated by wetting and drying on the

cells associated with more frequent and aggressive grid resolution coarsening at $\varepsilon = 10^{-3}$ than $10^{-4}$. Due to the smaller $\Delta t$ at $\varepsilon = 10^{-3}$, the trend in $N_{\Delta t}$ is steeper at $\varepsilon = 10^{-3}$ than $10^{-4}$, i.e. more timesteps are taken and thus more computational effort is spent by GPU-MWDG2 to reach a given simulation time at $\varepsilon = 10^{-3}$ than $10^{-4}$. Still, despite the steeper trend in $N_{\Delta t}$, the cumulative computational effort of performing the MRA process is similar using both $\varepsilon = 10^{-3}$ and $10^{-4}$, which is expected given that $R_{MRA}$ is also very similar for values of $\varepsilon$. In contrast, the cumulative computational effort of performing the DG2

solver updates is considerably higher at $\varepsilon = 10^{-4}$ than $10^{-3}$, likely due to the higher $R_{DG2}$ at $\varepsilon = 10^{-4}$, which seems correct since a higher computational effort to perform the DG2 solver updates per timestep should lead to a higher cumulative computational effort (assuming that the time histories of $N_{\Delta t}$ are similar for the different values of $\varepsilon$, which is the case here). Overall, for both values of $\varepsilon$, the total computational effort of running the GPU-MWDG2 simulations is always lower than that of the GPU-DG2 simulation, with $C_{tot}$ always being lower than that of GPU-DG2 at both $\varepsilon = 10^{-3}$ and $10^{-4}$. The

accumulated speedups of the GPU-MWDG2 simulations, $S_{acc}$, finish at around 2.5 and 2.0 using $\varepsilon = 10^{-3}$ and $10^{-4}$, respectively.

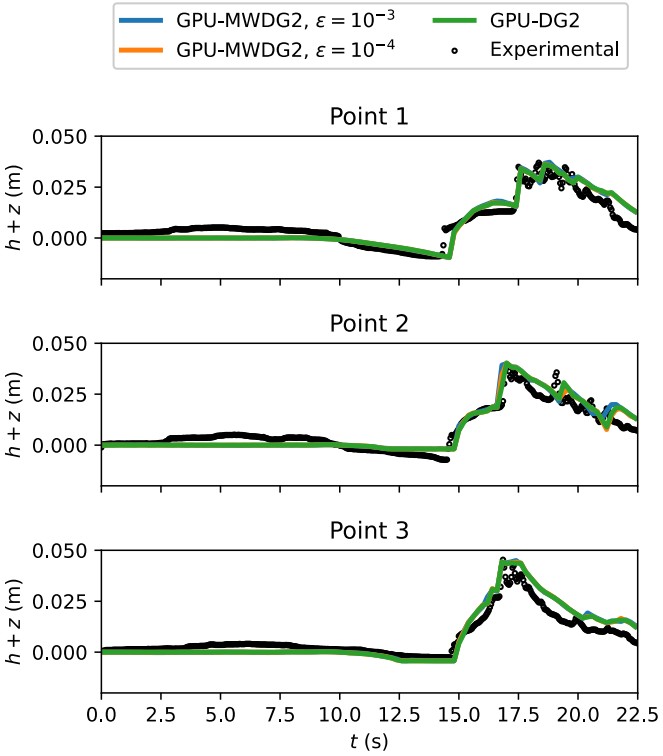

**Figure 8:** Monai Valley. Time series of the water surface elevation ($h + z$) predicted by GPU-DG2 and GPU-MWDG2 at the three sampling points (shown in Figure 6, top left panel) compared to the experimental results.


**Table 3:** Monai valley. RMSE and $r$ scores from the GPU-MWDG2 predictions versus the GPU-DG2 prediction.

| | RMSE | | $r$ | |
|---|---|---|---|---|
| **Prediction dataset** | $\varepsilon = 10^{-3}$ | $\varepsilon = 10^{-4}$ | $\varepsilon = 10^{-3}$ | $\varepsilon = 10^{-4}$ |
| Time series at Point 1 | $4.19 \times 10^{-4}$ | $1.36 \times 10^{-4}$ | 0.9995 | 0.9999 |
| Time series at Point 2 | $1.40 \times 10^{-3}$ | $7.55 \times 10^{-4}$ | 0.9935 | 0.9979 |
| Time series at Point 3 | $4.53 \times 10^{-4}$ | $1.91 \times 10^{-4}$ | 0.9994 | 0.9999 |
| Flood map at $t_{\text{end}}$ | $1.81 \times 10^{-3}$ | $1.00 \times 10^{-3}$ | 0.9978 | 0.9993 |

In Figure 8 and Table 3, the difference between the GPU-MWDG2 predictions at $\varepsilon = 10^{-3}$ and $10^{-4}$ and the GPU-DG2 prediction are evaluated for the time series of the water surface elevation at Points 1, 2 and 3 (coloured points in the top left

panel of Figure 7), at which all the predictions agree well with the measured time series. With both $\varepsilon$ values and at all three points, the GPU-MWDG2 predictions match those of GPU-DG2, yielding close RMSE and $r$ scores, including for the spatial flood map predictions at $t_{\text{end}}$. Overall, GPU-MWDG2 competitively reproduces the GPU-DG2 water surface elevation predictions with both $\varepsilon$ values while being more than 2 times faster than GPU-DG2 due to using $L = 10$, i.e. the "borderline" value of $L$ where GPU-MWDG2 adaptivity reliable yields a speedup. In the next test case, the impact of a larger DEM size,





requiring a larger $L$ value, on the speedup of the GPU-MWDG2 solver is evaluated, while further considering the prediction

of more complex velocity-related quantities.

### 3.2 Seaside Oregon

This is another popular benchmark test case used to validate hydrodynamic solvers for nearshore tsunami inundation

simulation (Gao et al., 2020; Macías et al., 2020; Park et al., 2013; Qin et al., 2018; Violeau et al., 2016). It involves a 1:50

scaled replica of an urban town in Seaside, Oregon, flooded by a tsunami travelling along a scaled DEM made up of 2181 ×

1091 cells, here requiring a larger $L$ = 12 to generate the initial *square uniform grid*.

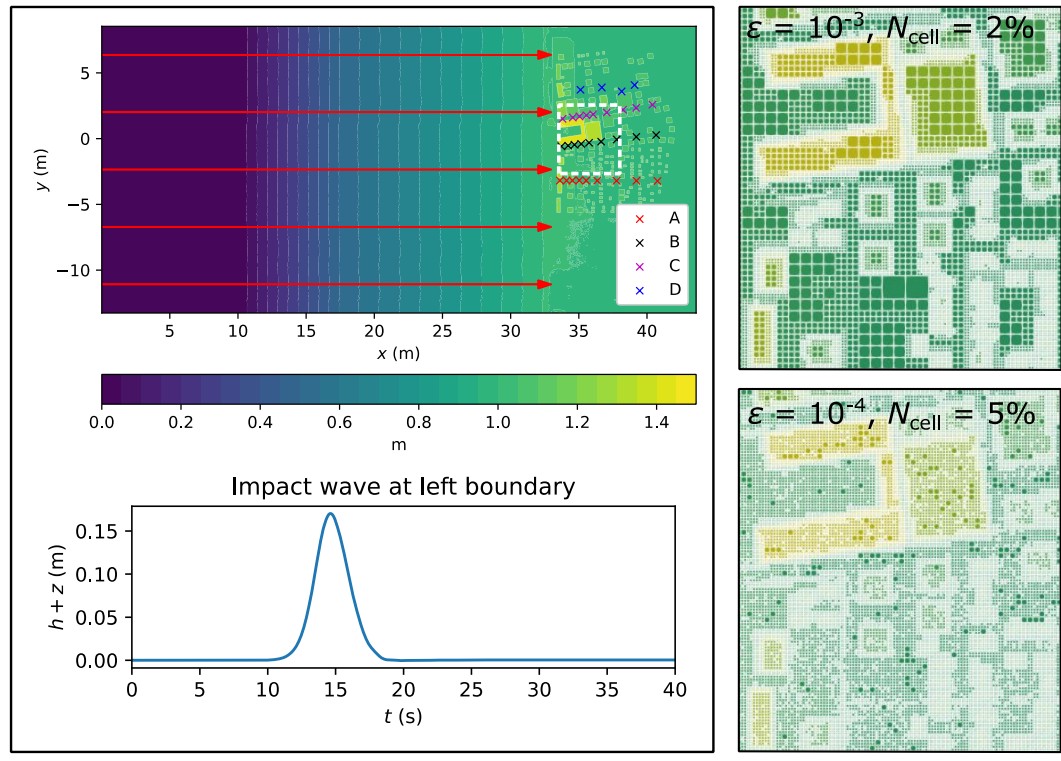

**Figure 9**: Seaside Oregon. Top-down view of bathymetry (top left panel), where the red arrows indicate the direction and
distance travelled; tsunami time history entering the left boundary (bottom left panel); initial GPU-MWDG2 grids (right
panels) for the potion in white box (top left panel).

In Figure 9, the bathymetric area is shown (top left panel), which is very plain everywhere except to the right where

very complex terrain features of the urban town, such as buildings and streets, are located. The urban town is flooded by a

tsunami that enters from the left boundary and travels a distance of 33 m to the right before hitting and flooding the town (as

shown by the red arrows). This tsunami is simulated for 40 s, during which it travels through the bathymetric area in four

stages of flow over time, much like in the last test case (Sect. 3.1): the entry stage (0 to 10 s), the travelling stage (10 to 25 s),

the flooding stage (25 to 35 s) and the inundation stage (35 to 40 s). During the entry stage, the tsunami is not yet in the





bathymetric area. During the travelling stage, the tsunami starts to enter the bathymetric area from the left boundary and travels right towards the town. During the flooding stage, the tsunami hits the town, flooding the streets and overtopping

some of the buildings, causing vigorous flow dynamics. Finally, during the inundation stage, the tsunami inundates the town and eventually interacts with the right boundary, causing wave reflections. The water surface elevation hydrograph of the tsunami is plotted in the bottom left panel of Figure 9: it only has a single peak, indicating low tsunami complexity. The right panels show the initial GPU-MWDG2 non-uniform grids at $\varepsilon = 10^{-3}$ and $10^{-4}$, respectively (again for the portion of the bathymetric area framed by the white box in top left panel): at $\varepsilon = 10^{-3}$, greater grid coarsening is achieved, with the grid

including only 2% of the number of cells as in the GPU-DG2 uniform grid, whereas at $\varepsilon = 10^{-4}$ it is 5% since more cells are used due to retention of finer resolution around and within complex terrain features of the urban town.

**Figure 10**: Seaside Oregon. Metrics of Table 1 applied to the GPU-MWDG2 and GPU-DG2 simulations. Also shown is a time history of $\Delta t$ (centre panel).





In Figure 10, an analysis of runtimes of the GPU-MWDG2 and GPU-DG2 simulations are shown. As indicated by the time history of $N_{cell}$, the number of cells in the GPU-MWDG2 non-uniform grid does not increase significantly for either value of $\varepsilon$ until the flooding stage of flow at 25 s, where $N_{cell}$ starts increasing more noticeably. Once the number of cells starts increasing, there is a corresponding increase in $R_{DG2}$. On the other hand, the time history of $R_{MRA}$ quite flat during the

entire simulation, except for a small decrease during the first 10 s of the simulation, and a small increasing trend in the final 5 s of the simulation, i.e. during the inundation stage of flow when the number of cells increases relatively sharply compared to the rest of the simulation. Driven primarily by the increase in $R_{DG2}$ at the flooding stage at 25 s, the time history of $S_{inst}$ is quite stable until 25 s and thereafter shows a decreasing trend that is particularly steep at $\varepsilon = 10^{-3}$.

Like the previous test case (Sect. 3.1), the time histories of $\Delta t$ in the GPU-DG2 simulation and GPU-MWDG2

simulation at $\varepsilon = 10^{-4}$ are very similar, but at $\varepsilon = 10^{-3}$, there is a sharp drop in $\Delta t$ after 32 s, i.e. when the flooding stage of flow starts transitioning to the inundation stage. This sharp drop in $\Delta t$ is triggered by wet/dry fronts at coarse cells that are present in the non-uniform grid with $\varepsilon = 10^{-3}$, but not with $10^{-4}$. The first drop in $\Delta t$, which occurs at 25 s when the flooding stage starts, leads to a locally steeper trend in the time history of $N_{\Delta t}$, as indicated by the kink at 25 s. The second drop in $\Delta t$, which is seen only for $\varepsilon = 10^{-3}$ after 32 s, leads to a sustained steepness in the time history of $N_{\Delta t}$ after 32 s. This steepness

means that GPU-MWDG2 takes more timesteps and thus accumulates more computational effort to reach a given simulation time at $\varepsilon = 10^{-3}$ than $10^{-4}$, which is confirmed by the final value of $C_{MRA}$, which is higher at the end of the simulation at $\varepsilon = 10^{-3}$ compared to $10^{-4}$, even though its time history at $\varepsilon = 10^{-3}$ was consistently lower than at $\varepsilon = 10^{-4}$ before this. Since $R_{MRA}$ was always lower at $\varepsilon = 10^{-3}$ than $10^{-4}$, this observation about $C_{MRA}$ suggests that even if the computational effort per timestep is lower throughout the simulation, a high timestep count can sufficiently increase the cumulative computational

effort such that it becomes higher at $\varepsilon = 10^{-3}$ than $10^{-4}$. Nonetheless, the time history of $C_{tot}$ in the GPU-MWDG2 simulations always remains well below that of the GPU-DG2 simulation, with $S_{inst}$ finishing at 3.5 and 3.0 with $\varepsilon = 10^{-3}$ and $10^{-4}$, respectively.

In Figure 11 and Table 4, the difference between the GPU-MWDG2 predictions at $\varepsilon = 10^{-3}$ and $10^{-4}$ and the GPU-DG2 prediction are evaluated in terms of time series of the water surface elevation, $h + z$, the $u$ component of the velocity

field, and the associated momentum, $M_x = 0.5hu^2$, at points A1 (one of the left-most crosses in Figure 9), B6 (one of the central crosses) and D4 (one of the right-most crosses), all showing a good agreement with the measured time series (also plotted in Figure 11). Point A1 is located at the bottom left corner at the start of the town, at which GPU-MWDG2 closely trails the GPU-DG2 predictions for both $\varepsilon$ values ($r$ scores of 0.99 and the same order-of-magnitude for the RMSE scores). Point B6 is located in the middle of the urban town, at which the RMSE and $r$ scores are similar to those obtained at point

A1; however, GPU-MWDG2 at $\varepsilon = 10^{-3}$ provides improved visual trailing of the GPU-DG2 predicted velocity. Point D4 is located at the downstream end of the upper area in the urban town, at which less similarity is detected between the GPU-MWDG2 predictions and the GPU-DG2 prediction for $u$ and $M_x$: at $\varepsilon = 10^{-3}$, lower $r$ scores of around 0.88 and 0.77 are detected, respectively, compared to the $r$ scores of around 0.91 and 0.92 reached at $\varepsilon = 10^{-4}$, for which the $u$ velocity is more





closely predicted (RSME score of $9.51 \times 10^{-2}$ at $\varepsilon = 10^{-4}$ versus a score of $1.30 \times 10^{-1}$ at $\varepsilon = 10^{-3}$). In terms of spatial flood

map at $t_{\text{end}}$, the same discrepancies between the $r$ scores can be seen for the $u$ velocity predictions, and between the RSME

score for the $M_x$ predictions (Table 4). Hence, $\varepsilon = 10^{-4}$ can be a better choice to acquire more accurate velocity-related

predictions in the zones inside and around fine-scale terrain features of urban town, while $\varepsilon = 10^{-3}$ remains a competitive

choice to maximise the speedup throughout the simulation by an order-of-magnitude. This and the previous test cases (Sects.

3.1 and 3.2) show that the GPU-MWDG2 solver can achieve at least a 2-fold speedup over the GPU-DG2 solver for tsunami

simulations involving a single-wave impact event. In the following Sects. 3.3 and 3.4, the speedup of the GPU-MWDG2

solver is evaluated for field-scale scenarios involving more complex tsunami impact events.

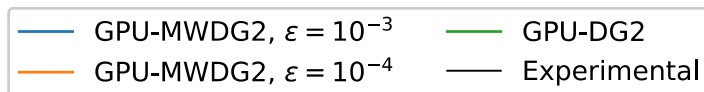

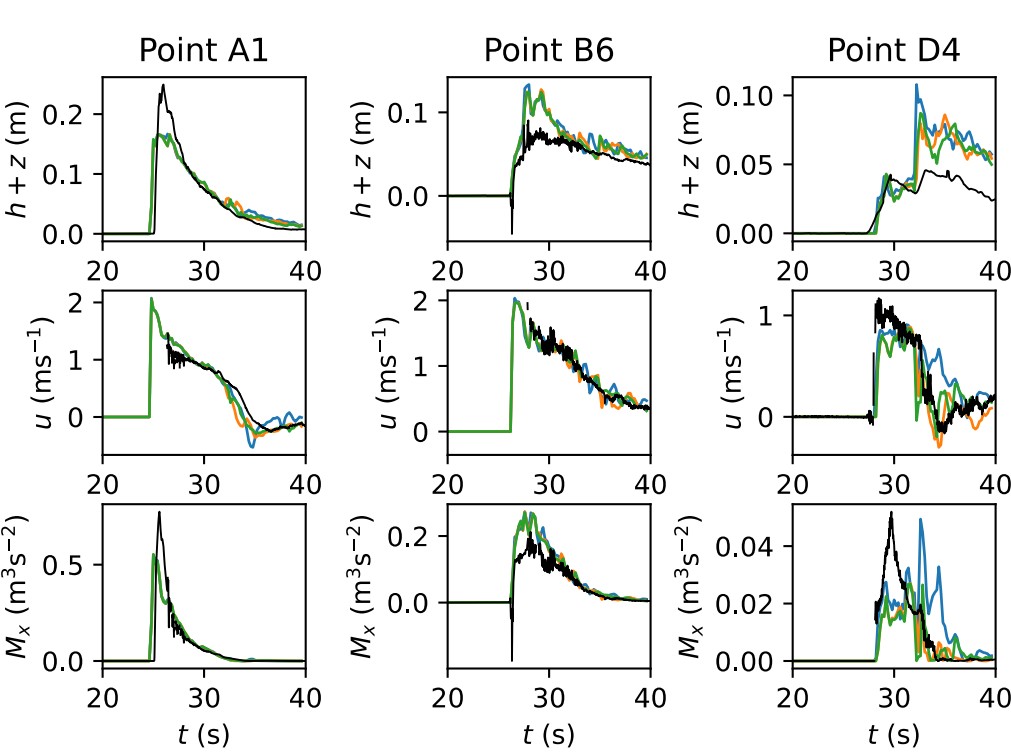

**Figure 11**: Seaside Oregon. Time series of the water surface elevation ($h + z$), $u$ velocity component and momentum $M_x$ for the GPU-DG2 and GPU-MWDG2 predictions at points A1, B6 and D4 (Figure 9), compared to the experimental data.






**Table 4**: Seaside Oregon. RMSE and *r* scores from GPU-MWDG2 predictions versus the GPU-DG2 prediction.

| Prediction dataset | Quantity | RMSE | | *r* | |
|---|---|---|---|---|---|
| | | $\varepsilon = 10^{-3}$ | $\varepsilon = 10^{-4}$ | $\varepsilon = 10^{-3}$ | $\varepsilon = 10^{-4}$ |
| **Time series at A1** | $h + z$ | $3.04 \times 10^{-3}$ | $2.68 \times 10^{-3}$ | 0.9979 | 0.9982 |
| | $u$ | $6.86 \times 10^{-2}$ | $4.54 \times 10^{-2}$ | 0.9905 | 0.9958 |
| | $M_x$ | $2.90 \times 10^{-3}$ | $1.01 \times 10^{-3}$ | 0.9995 | 0.9999 |
| **Time series at B6** | $h + z$ | $4.48 \times 10^{-3}$ | $2.46 \times 10^{-3}$ | 0.9931 | 0.9976 |
| | $u$ | $5.65 \times 10^{-2}$ | $4.29 \times 10^{-2}$ | 0.9944 | 0.9968 |
| | $M_x$ | $7.40 \times 10^{-3}$ | $3.87 \times 10^{-3}$ | 0.9943 | 0.9982 |
| **Time series at D4** | $h + z$ | $5.87 \times 10^{-3}$ | $3.75 \times 10^{-3}$ | 0.9841 | 0.9897 |
| | $u$ | $1.30 \times 10^{-1}$ | $9.51 \times 10^{-2}$ | 0.8889 | 0.9176 |
| | $M_x$ | $6.52 \times 10^{-3}$ | $2.20 \times 10^{-3}$ | 0.7746 | 0.9214 |
| **Spatial map at $t_{\text{end}}$** | $h + z$ | $5.58 \times 10^{-3}$ | $1.50 \times 10^{-3}$ | 0.9999 | 0.9999 |
| | $u$ | $6.66 \times 10^{-2}$ | $3.52 \times 10^{-2}$ | 0.8303 | 0.9502 |
| | $M_x$ | $2.03 \times 10^{-3}$ | $5.89 \times 10^{-4}$ | 0.9350 | 0.9945 |

### 3.3 Tauranga Harbour

This test case reproduces the 2011 Japan tsunami event in Tauranga Harbour, New Zealand (Borrero et al., 2015; Macías et al., 2015; 2020). The bathymetric area has a DEM made of $4096 \times 2196$ cells, requiring $L = 12$ to generate the initial *square uniform grid*. As shown by the red arrows in Figure 12, the tsunami enters from the top boundary and travels a short distance downwards before quickly hitting the coast at $y = 16$ km. As shown by the time history of the water surface elevation (bottom left panel of Figure 12), the tsunami is a wave train made up of three wave peaks and troughs that enter that bathymetric area one after the other at 0, 12 and 24 hr during the 40-hr tsunami event, with the latter two waves also exhibiting noise. Due to the wave train, the short travel distance before flooding the harbour, and the highly irregular bathymetric zones that trigger wave reflections and diffractions, vigorous flow dynamics occur within the bathymetric area from the very beginning of the simulation. The right panels of Figure 6 include the GPU-MWDG2 grids generated at $\varepsilon = 10^{-3}$ and $10^{-4}$ for the region bounded by the white box (top left panel). With $\varepsilon = 10^{-3}$, the number of cells in the grid is 5% of the



GPU-DG2 uniform grid, whereas with $\varepsilon = 10^{-4}$, it is 15%, due to less coarsening in and around the irregular bathymetric zones.

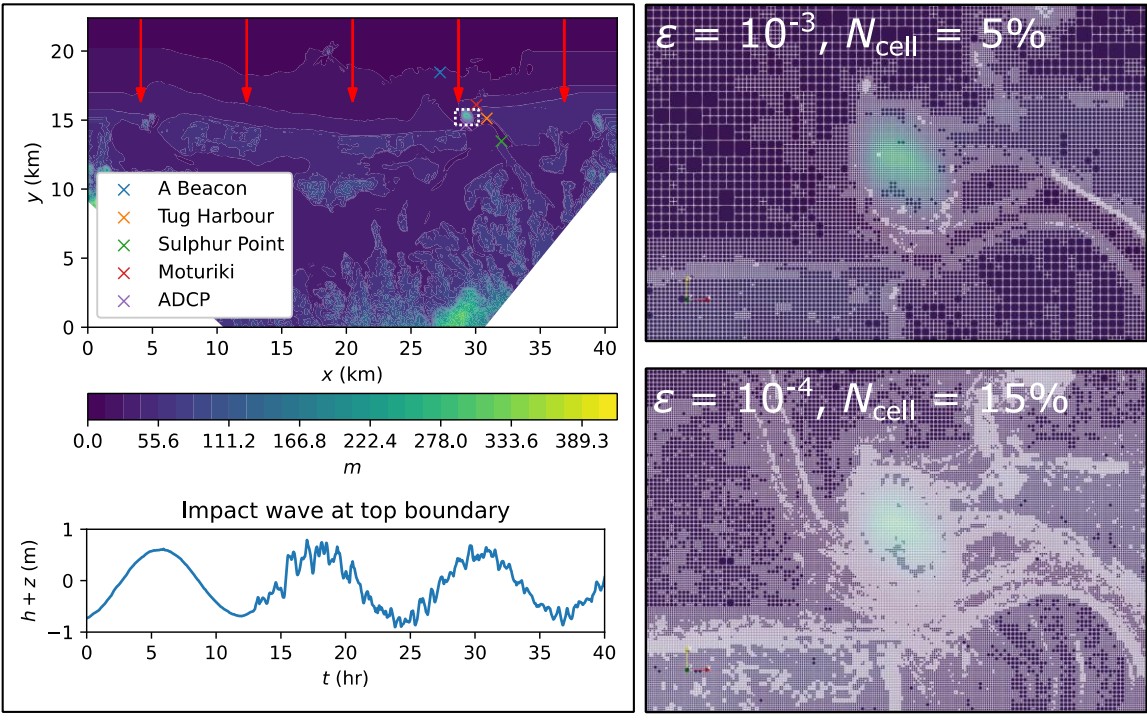

**Figure 12**: Tauranga Harbour. Top-down view of bathymetry (top left panel), where the red arrows indicate the direction and distance travelled; tsunami time history entering the top boundary (bottom left panel); initial GPU-MWDG2 grids (right panels) for the potion in white box (top left panel).

In Figure 13, an analysis of the runtimes of the GPU-MWDG2 and GPU-DG2 simulations is shown. Unlike the previous test cases (Sects. 3.1 and 3.2), the first wave of the tsunami wave train enters the bathymetric immediately, causing $N_{cell}$ to increase very sharply and immediately from its the initial value for both values of $\varepsilon$, which thereafter fluctuates due to the periodic tsunami signal. Following the sharp increase and fluctuations in $N_{cell}$, $R_{DG2}$ also sharply increases and fluctuates. However, $R_{MRA}$ does not and stays stable and flat throughout the simulation. Thus, driven primarily by the sharp decrease in $R_{DG2}$, $S_{inst}$ decreases sharply from 4.0 to 1.6. The time histories of $\Delta t$ in the GPU-DG2 simulation and the GPU-MWDG2 simulation using $\varepsilon = 10^{-4}$ follow each other quite closely, but at $\varepsilon = 10^{-3}$, the time history of $\Delta t$ shows two periodic drops after 24 h, likely due to periodic wetting and drying processes around coarse cells that are present in the non-uniform grid at $\varepsilon = 10^{-3}$ but not at $10^{-4}$. Due to the smaller $\Delta t$ at $\varepsilon = 10^{-3}$, the time history of $N_{\Delta t}$ is locally steeper (see the kinks at 25 and 35 h), but this does not lead to significant differences between the cumulative computational effort at $\varepsilon = 10^{-3}$ versus $10^{-4}$. The time history of $C_{tot}$ in the GPU-MWDG2 simulations consistently remains below that of the GPU-DG2 simulation for both values of $\varepsilon$, but they are relatively close to each other compared to the previous test cases (Sects. 3.1 and 3.2). Thus,



even though $S_{acc}$ starts at around 4, like in the previous test case with $L = 12$ (Sect. 3.2), it drops sharply to 1.6 and 1.4 at $\varepsilon = 10^{-3}$ and $10^{-4}$, respectively.



**Figure 13**: Tauranga Harbour. Metrics of Table 1 applied to the GPU-MWDG2 and GPU-DG2 simulations. Also shown is a time history of $\Delta t$ (centre panel).

In Figure 14 and Table 5, the difference between the GPU-MWDG2 predictions at $\varepsilon = 10^{-3}$ and $10^{-4}$ and the GPU-DG2 prediction is evaluated for the time series of water surface elevation at sampling points A Beacon, Tug Harbour,
Sulphur Point and Moturiki (top left panel of Figure 12), all of which show a good agreement with the measured time series. The GPU-MWDG2 predicted water surface elevations are very close to those predicted by GPU-DG2 regardless of $\varepsilon$, as confirmed by the $r$ scores, around 0.99, and by the RSME scores, which remain in the same order-of-magnitude (Table 5).



Moreover, the difference for the $Speed = \sqrt{u^2 + v^2}$ time series at point ADCP (top left panel of Figure 12) is evaluated, showing less agreement with the measured time series compared to the water surface elevation: the GPU-MWDG2

predictions at $\varepsilon = 10^{-4}$ shows 5% better similarity to the GPU-DG2 predictions compared to at $\varepsilon = 10^{-3}$ ($r$ scores of 0.9527 and 0.9023, respectively). However, the better similarity score at $\varepsilon = 10^{-4}$ is mostly detectable in the prediction of the final flood map at $t_{end}$ leading to a 15% higher $r$ score of 0.89 compared to the score of 0.75 at $\varepsilon = 10^{-3}$ (Table 5).

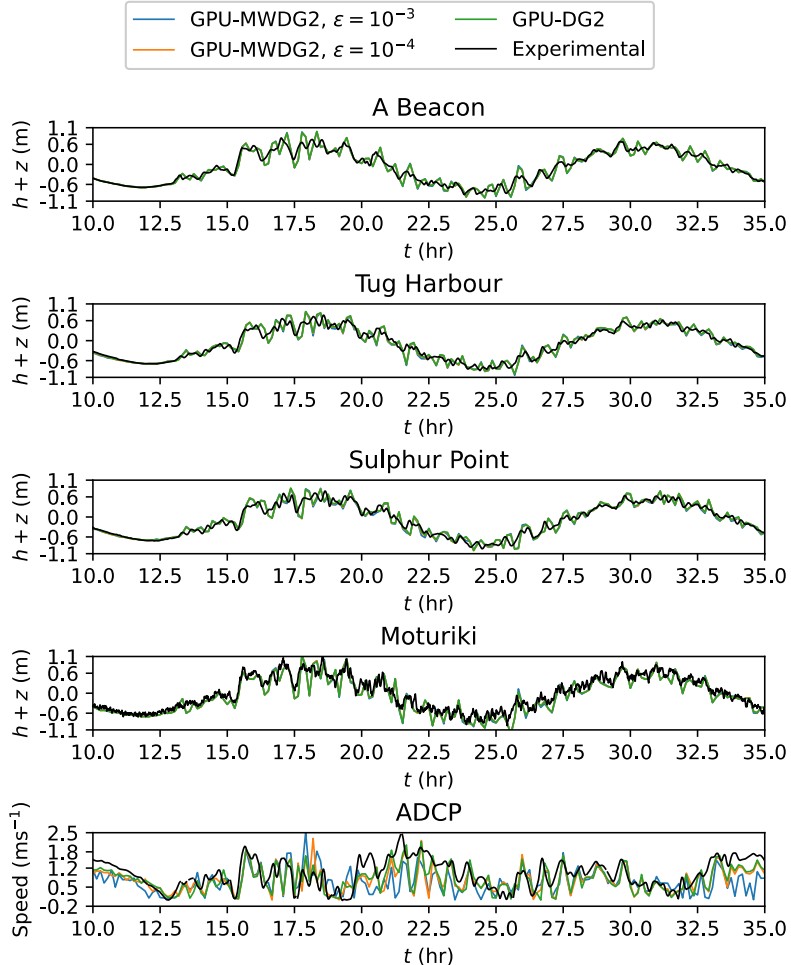

**Figure 14**: Tauranga Harbour. Time series of the water surface elevation ($h + z$) produced by GPU-DG2 and GPU-MWDG2
at the points labelled A Beacon, Tug Harbour, Sulphur Point and Moturiki (labelled in Figure 12) and for the $Speed$, at the point ADCP (also labelled in Figure 12), compared to the experimental results.

Overall, this test case features a more complex tsunami compared to the previous test cases (Sects. 3.1 and 3.2), which sharply increases the number of cells in the GPU-MWDG2 non-uniform grid and thus also increases the

computational effort of performing the DG2 solver updates. Hence, despite requiring the same $L = 12$ as the previous test case (Sect. 3.2), the final speedups are lower in this test case due to the more complex tsunami, with $S_{acc}$ finishing at 1.6-





and 1.4-fold with $\varepsilon = 10^{-3}$ and $10^{-4}$, respectively. Using $\varepsilon = 10^{-4}$ would improve the closeness to the GPU-DG2 predicted velocities, while using $\varepsilon = 10^{-3}$ leads to very close water surface elevation predictions and fairly accurate velocity predictions, although without a major improvement in the speedup. In the next test case, another complex tsunami with higher frequency impact event peaks is considered, but now with a smaller DEM size requiring $L = 10$.

**Table 5**: Tauranga Harbour. RMSE and $r$ scores for GPU-MWDG2 predictions versus the GPU-DG2 prediction.

| | | RMSE | | $r$ | |
|---|---|---|---|---|---|
| **Prediction dataset** | **Quantity** | $\varepsilon = 10^{-3}$ | $\varepsilon = 10^{-4}$ | $\varepsilon = 10^{-3}$ | $\varepsilon = 10^{-4}$ |
| Time series at A Beacon | $h + z$ | $7.46 \times 10^{-2}$ | $1.28 \times 10^{-2}$ | 0.9986 | 0.9999 |
| Time series at Tug Harbour | $h + z$ | $6.86 \times 10^{-2}$ | $8.99 \times 10^{-2}$ | 0.9984 | 0.9972 |
| Time series at Sulphur Point | $h + z$ | $1.62 \times 10^{-1}$ | $1.43 \times 10^{-1}$ | 0.9916 | 0.9933 |
| Time series at Moturiki | $h + z$ | $7.81 \times 10^{-2}$ | $2.35 \times 10^{-2}$ | 0.9935 | 0.9993 |
| Time series at ACDP | $Speed$ | $3.45 \times 10^{-1}$ | $2.40 \times 10^{-1}$ | 0.9023 | 0.9527 |
| Spatial map at $t_{\text{end}}$ | $h + z$ | $7.33 \times 10^{-2}$ | $3.10 \times 10^{-2}$ | 0.9999 | 0.9999 |
| | $Speed$ | $8.01 \times 10^{-2}$ | $4.80 \times 10^{-2}$ | 0.7556 | 0.8950 |

### 3.4 Hilo Harbour

This test case reproduces the 2011 Japan tsunami event at Hilo Harbour in Hawaii, USA (Arcos & LeVeque, 2014; Lynett et al., 2017; Macías et al., 2020; Velioglu-Sogut & Yalciner, 2019). It involves a complex tsunami made up of a high-frequency wave train that propagates for 6 hr into a bathymetric area that is smaller than the previous test case (Sect. 3.3). The latter bathymetric area has a DEM size made of $702 \times 692$ cells, requiring a smaller $L = 10$ to generate the initial *square uniform grid*. As shown in Figure 15, the tsunami enters from the top boundary and travels south to flood and interact with the coast at $y = 4$ km. The wave train occurs over the entire 6 hr simulation time, from a reference timestamp of 7 hr to 13 hr post-earthquake. In Figure 15, the time history of the wave train is shown during the 8.5 to 11 hr time period (bottom left panel): from the very beginning and during the entire simulation, violent flow dynamics occur in the bathymetric area. The right panels show the initial GPU-MWDG2 non-uniform grids generated at $\varepsilon = 10^{-3}$ and $10^{-4}$ for the region bounded by the white box (top left panel): the number of cells in the initial non-uniform grids are at 10.0% and 22.5% of the GPU-DG2 uniform grid at $\varepsilon = 10^{-3}$ and $\varepsilon = 10^{-4}$, respectively.



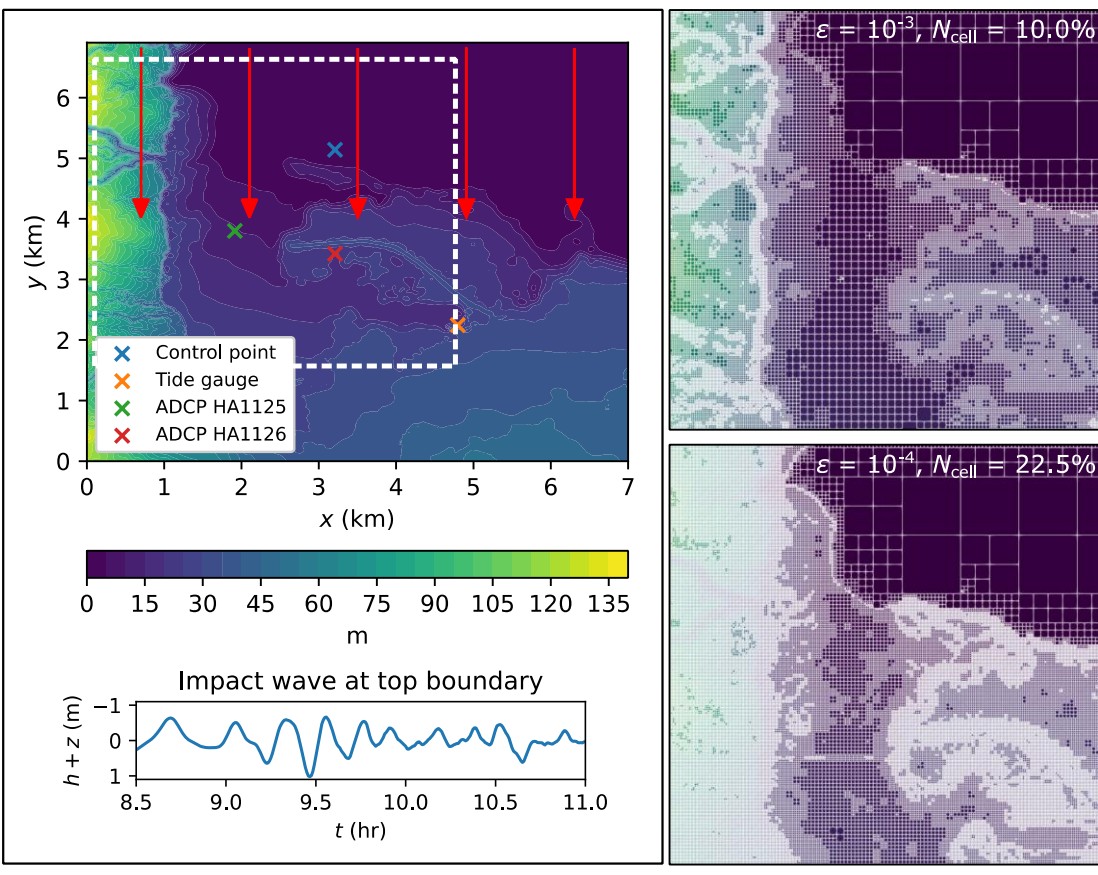

**Figure 15:** Hilo Harbour test case. Top-down view of bathymetry (top left panel), where the red arrows indicate the direction and distance travelled; tsunami time history entering the top boundary (bottom left panel); initial GPU-MWDG2 grids (right panels) for the potion in white box (top left panel).

In Figure 16, an analysis of runtimes of the GPU-MWDG2 and GPU-DG2 simulations is shown. Like the last test case (Sect. 3.3), since the wave train enters the bathymetric area immediately, $N_{cell}$ increases sharply and immediately to maximum values of 32% and 40% at $\varepsilon = 10^{-3}$ and $10^{-4}$, respectively. The time history of $N_{cell}$ stays at this maximum for the simulation except for (somewhat less sharp) localised drops at certain simulation times, e.g. at 8 hr. Following $N_{cell}$, $R_{DG2}$ also increases sharply and immediately (to maximum values of 50% and 55% at $\varepsilon = 10^{-3}$ and $10^{-4}$, respectively), and shows localised drops at the same timestamps as the drops in $N_{cell}$. In contrast, the time history of $R_{MRA}$ stays very flat throughout the simulation, except for a small, temporary drop at 8 hr, which is when the largest drop in $N_{cell}$ occurs. Due to the generally flat time histories of $R_{DG2}$ and $R_{MRA}$, the time history of $S_{inst}$ is also flat except for localised peaks that occur at the same timestamps as the localised drops in $R_{DG2}$ and $R_{MRA}$. Unlike all of the previous test cases (Sect. 3.1 - 3.3), the time history of $\Delta t$ is very similar between the GPU-DG2 simulation and the GPU-MWDG2 simulations, regardless of the $\varepsilon$ value as they both yield high number of cells compared to the uniform GPU-DG2 grid. Thus, the time history of $N_{\Delta t}$ is virtually identical across all simulations. Given that the time histories of $N_{\Delta t}$, $R_{DG2}$ and $R_{MRA}$ are similar for both $\varepsilon$ values, the time





histories of $C_{DG2}$ and $C_{MRA}$ are also very similar. The time history of $C_{tot}$ is very close between the GPU-DG2 simulation and the GPU-MWDG2 simulations in this test case (even more so than in the last case, Sect. 3.3), so $S_{acc}$ is the lowest out of

all the test cases, finishing at 1.25 and 1.10 using $\varepsilon = 10^{-3}$ and $10^{-4}$, respectively.

**Figure 16**: Hilo Harbour. Metrics of Table 1 applied to the GPU-MWDG2 and GPU-DG2 simulations. Also shown is a time history of $\Delta t$ (centre panel).

In Figure 17 and Table 6, the difference between the GPU-MWDG2 predictions at $\varepsilon = 10^{-3}$ and $10^{-4}$ and the GPU-DG2 prediction is evaluated in terms of the time series of the water surface elevation at points labelled "Control point" and "Tide gauge", and $u$ and $v$ velocity components at the points labelled "ADCP HA1125" and "ADCP HA1126" (top left panel in Figure 15), all showing a fair agreement with the measured time series. For both $\varepsilon$, the GPU-MWDG2 predicted water





surface elevations are more than 97% similar to those predicted by GPU-DG2 (Table 6), with broadly comparable closeness

for the RMSEs that tends to improve at $\varepsilon = 10^{-4}$ for the spatial map at $t_{end}$ and the time series at "Control point". In terms of the GPU-MWDG2 predicted velocities, the closeness is comparable in terms of the RMSE scores, but the $r$ scores can vary by 10% depending on the choice of $\varepsilon$: $\varepsilon = 10^{-4}$ yield betters $r$ scores between around 0.82 and 0.91, whereas the scores yielded at $\varepsilon = 10^{-3}$ varied between around 0.70 and 0.83 – the highest discrepancies occurred in the spatial flood maps at $t_{end}$ (Table 6).


**Table 6**: Hilo Harbour. RMSE and $r$ scores for GPU-MWDG2 predictions versus the GPU-DG2 prediction.

| Prediction dataset | Quantity | RMSE | | $r$ | |
|---|---|---|---|---|---|
| | | $\varepsilon = 10^{-3}$ | $\varepsilon = 10^{-4}$ | $\varepsilon = 10^{-3}$ | $\varepsilon = 10^{-4}$ |
| Control point | $h + z$ | $1.11 \times 10^{-1}$ | $5.45 \times 10^{-2}$ | 0.9767 | 0.9923 |
| Tide gauge | $h + z$ | $1.81 \times 10^{-1}$ | $1.10 \times 10^{-1}$ | 0.9827 | 0.9938 |
| ADCP HA1125 | $v$ | $3.63 \times 10^{-1}$ | $2.91 \times 10^{-1}$ | 0.7884 | 0.8252 |
| ADCP HA1126 | $u$ | $3.23 \times 10^{-1}$ | $2.23 \times 10^{-1}$ | 0.8343 | 0.9193 |
| Spatial map at $t_{end}$ | $h + z$ | $5.31 \times 10^{-2}$ | $6.20 \times 10^{-3}$ | 0.9999 | 0.9999 |
| | $v$ | $9.17 \times 10^{-2}$ | $6.09 \times 10^{-2}$ | 0.7063 | 0.8134 |
| | $u$ | $9.01 \times 10^{-2}$ | $5.93 \times 10^{-2}$ | 0.7465 | 0.8387 |

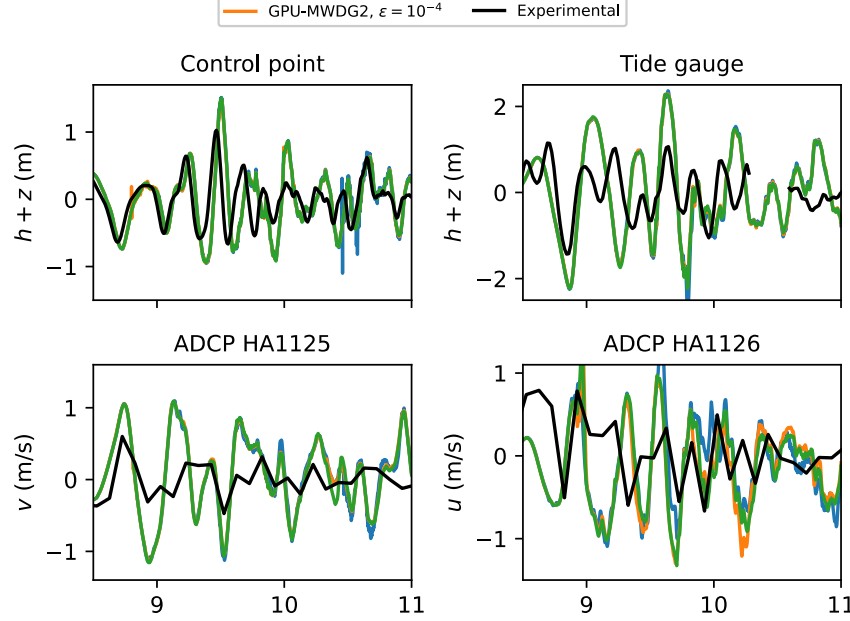

**Figure 17**: Hilo Harbour. Time series produced by GPU-DG2 and GPU-MWDG2, for the water surface elevation ($h + z$) at "control point" and "Tide gauge" (labelled in Figure 17), and for the velocity components at "ADCP HA1125" and ADCP
HA 1126 (labelled in Figure 17), compared to the experimental results.





Overall, despite simulating a test case with a complex tsunami impact event and also a DEM size that requires selecting a small $L = 10$, GPU-MWDG2 still manages to attain speedups over GPU-DG2 in this test case: around 1.25 at $\varepsilon = 10^{-3}$ and 1.10 at $\varepsilon = 10^{-4}$. This seems to suggest that the GPU-MWDG2 solver can reliably be used to gain speedups over the GPU-DG2 solver even if simulating complex tsunami impact events, using $\varepsilon = 10^{-3}$ to boost the speedup, or using $\varepsilon = 10^{-4}$ increase the quality of velocity predictions.

## 4 Conclusions and recommendations

This work reported the version release of the LISFLOOD-FP 8.2 hydrodynamic modelling framework, which integrates the GPU parallelised grid resolution adaptivity of multiwavelets (MW) within the second-order discontinuous Galerkin (DG2) solver of the shallow water equations (GPU-MWDG2) to run simulations on a non-uniform grid. The GPU-MWDG2 solver enables dynamic (in time) grid resolution adaptivity based on both the (time-varying) flow solutions and the (time-invariant) Digital Elevation Model (DEM) representations. It has been aimed at reducing the runtime of the existing uniform grid GPU parallelised DG2 solver (GPU-DG2) for flood simulation driven by rapid, multiscale impact events, which were exemplified by tsunami-induced flooding.

The framework integrating dynamic GPU-MWDG2 adaptivity in LISFLOOD-FP 8.2 was reported with a focus on: how to run GPU-MWDG2 simulations from raster-formatted DEM and initial flow condition setup files, requiring a user-specified maximum refinement level, $L$, and an error threshold, $\varepsilon$; consideration of the memory limits affordable per selected $L$ and per GPU card; and, the development of a suite of time-dependent metrics for assessing the potential speedup afforded by GPU-MWDG2 adaptivity. The accuracy and efficiency of dynamic GPU-MWDG2 adaptivity was assessed for tsunami-induced flood simulations featuring different levels of impact event complexity, ranging from a single-wave tsunami to a wave train of multiple tsunamis. The assessments qualitatively and quantitatively evaluated the capability of dynamic GPU-MWDG2 adaptivity, using $\varepsilon = 10^{-3}$ and $10^{-4}$, in reproducing spatial and temporal GPU-DG2 predictions of water levels and velocity-related quantities. The evaluations consistently demonstrated that the GPU-MWDG2 simulations using $\varepsilon = 10^{-3}$ yield water level predictions as accurate as the GPU-DG2 simulations, and that using the smaller $\varepsilon = 10^{-4}$ would only be a potential option to improve the accuracy of velocity-related predictions if needed.

In terms of the average speedup that can be achieved by dynamic GPU-MWDG2 adaptivity, it seems to be maximised depending on whether: (i) the size and resolution of the DEM area corresponds to a choice for $L \geq 9$ and (ii) the simulated impact event is single-peaked, such as a single-wave tsunami. As shown in Table 7, for the impact events that are single-peaked: when the DEM area required $L = 10$, the average speedups would be around 2.0 times faster than the GPU-DG2 simulations (i.e. "Monai Valley"); whereas, with the DEM area requiring a larger $L = 12$, considerable average speedups of 3.3-fold were achieved at $\varepsilon = 10^{-4}$, which increased to 4.5-fold at $\varepsilon = 10^{-3}$ (i.e. "Seaside, Oregon"). Meanwhile, for the multi-peaked impact events, the average speedups reduced to 1.8-fold for a DEM area requiring $L = 12$ (i.e. "Tauranga Harbour") and to 1.2-fold for a smaller DEM requiring a smaller $L = 10$ (i.e. "Hilo Harbour").





**Table 7**: Summary of GPU-MWDG2 runtimes and potential average speedups with respect to GPU-DG2.

| | | | GPU-MWDG2 | | | | GPU-DG2 |
|---|---|---|---|---|---|---|---|
| **Tsunami (impact) event** | | **$L$ (Max cells)** | **Runtime, $\varepsilon$** | | **Speedup, $\varepsilon$** | | **Runtime** |
| **Test case** | $t_{end}$ | Single-wave tsunami | ---- | $10^{-3}$ | $10^{-4}$ | $10^{-3}$ | $10^{-4}$ | ---- |
| Monai Valley | 22.5 (9000 s*) | Yes | 10 (> 1.04m) | 16 s | 20 s | 2.5 | 2.0 | 40 s |
| Seaside Oregon | 40 s (33 min*) | Yes | 12 (> 16.7m) | 3.5 min | 5.2 min | 4.5 | 3.3 | 13 min** |
| Tauranga Harbour | 40 hr | No (three peaks) | 12 (> 16.7m) | 7.5 hr | 8.1 hr | 1.8 | 1.4 | 11.3 hr |
| Hilo Harbour | 6 hr | No (eleven peaks) | 10 (> 1.04m) | 5.3 min | 5.8 min | 1.3 | 1.2 | 6.9 min |

\* By accounting for the physical scaling factor of the replica.

In summary, the GPU-MWDG2 solver in LISFLOOD-FP 8.2 accelerates GPU-DG2 simulations of rapid multiscale flooding flows, yielding the greatest speedups for simulations needed $L \geq 10$ (i.e., ratio the DEM area to DEM resolution)

and driven by single-peaked impact events. The LISFLOOD-FP 8.2 code is accessible on Zenodo, DOI: 10.5281/zenodo.4073010, together with the input files and scripts for reproducing the simulation data, DOI: 10.5281/zenodo.13909072, with step-by-step guidance at https://www.seamlesswave.com/Adaptive (last accessed: 9 October 2024).



*Code and data availability.* LISFLOOD-FP 8.2 source code is available from Zenodo (LISFLOOD-FP developers, 2024; https://zenodo.org/doi/10.5281/zenodo.4073010) as well as the simulation data and input files and scripts for reproducing them (Chowdhury, 2024; https://doi/org/10.5281/zenodo.13909072).

*Video supplement.* Step-by-step instructions on how to download and install and run the LISFLOOD-FP-8.2 code for the "Hilo Harbour" (Sect. 3.4) is available from Zenodo (Chowdhury, 2024). The video demo also includes updates on the changes made to the CMake build process for compatibility with different versions of the CUDA toolkit.

*Author contributions.* AAC coded, optimised and integrated the GPU-MWDG2 into the LISFLOOD-FP framework (methodology; software; validation; investigation; data curation; visualization; formal analysis). GK contributed to the conceptualisation, formal analysis, funding acquisition and project administration. Both AAC and GK conceived and wrote the paper (original draft; review and editing).

*Competing interests.* The contact author has declared that none of the authors has any competing interests.

*Acknowledgements.* The authors are extremely grateful for Charles Rougé from the University of Sheffield for his feedback on the efficiency analysis of Sect. 3, and Paul Bates and Jefferey Neal from the University of Bristol for their support of the collaborative open source LISFLOOD-FP project.

*Financial Support.* AAC and GK were supported by the UK Engineering and Physical Sciences Research Council (EPSRC) grant EP/R007349/1.

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

**Appendix A: The GPU-MWDG2 algorithm**

The GPU-MWDG2 algorithm that is integrated into LISFLOOD-FP 8.2 solves the two-dimensional shallow water equations over a non-uniform grid that locally adapts its grid resolution to the flow solutions and DEM representation every simulation timestep. The conservative form of the shallow water equations in vectorial format is as follows:

$$\partial_t \boldsymbol{U} + \partial_x \boldsymbol{F}(\boldsymbol{U}) + \partial_y \boldsymbol{G}(\boldsymbol{U}) = \boldsymbol{S}_b(\boldsymbol{U}) + \boldsymbol{S}_f(\boldsymbol{U}) \tag{A1}$$

Where $\partial_{\square}$ represents a partial derivative operator; $\boldsymbol{U} = [h, hu, hv]^T$ is the vector of the flow variables where $T$ stands for the transpose operator, $h(x, y, t)$ is the water depth (m) at time $t$ and location $(x, y)$, and $u(x, y, t)$ and $v(x, y, t)$ are the $x$- and $y$-component of the velocity field (m/s) in the two-dimensional Cartesian space; $\boldsymbol{F} = [hu, (hu)^2 h^{-1} + 0.5\, gh^2, huv]^T$ and $\boldsymbol{G} = [hv, huv, (hv)^2 h^{-1} + 0.5gh^2]^T$ are the components of the flux vector in which $g$ is the gravitational acceleration constant (m/s²); $\boldsymbol{S}_b = [0, -gh\partial_x z, -gh\partial_y z]^T$ is the bed-slope source term vector incorporating the partial derivative of the bed elevation function $z(x, y)$; and $\boldsymbol{S}_f = [0, -C_f u\sqrt{u^2 + v^2}, -C_f v\sqrt{u^2 + v^2}]^T$ is the friction source term vector including the friction effects as function of $C_f = gn_M^2\, h^{-1/3}$ in which $n_M$ is Manning's roughness parameter. For ease of presentation, the scalar variable $s$ will hereafter be used to represent any of the physical flow quantities in $\boldsymbol{U}$ as well as the bed elevation $z$.

Over each computational cell $c$, the DG2 modelled data for any of the any physical flow quantities, $s \in \{h, hu, hv\}$, follows a piecewise-planar solution, denoted by $s_c(x, y, t)$ (Kesserwani & Sharifian, 2020). The piecewise-planar solution, $s_c(x, y, t)$, is expanded onto local basis functions from the scaled and truncated Legendre basis (Kesserwani et al., 2018;





Kesserwani & Sharifian, 2020) to become spanned by three shape coefficients: $\mathbf{s_c} = [s_c^0(t), s_c^{1x}(t), s_c^{1y}(t)]^T$, where $s_c^0(t)$ is a coefficient of an average; and $s_c^{1x}(t)$ and $s_c^{1y}(t)$ are $x$- and $y$-directional slope coefficients, respectively (see Eq. 10 in Kesserwani & Sharifian, 2020). The bed-elevation, $s \in \{z\}$, is also represented as piecewise-planar, $s_c(x, y)$, but it is

spanned by time-independent shape coefficients. The shape coefficients in $\mathbf{s}_c$, $s \in \{h, hu, hv, z\}$, must be initialised (Eq. 11 in Kesserwani & Sharifian, 2020), while the time-dependent ones, with $s \in \{h, hu, hv\}$, are updated using "DG2 solver updates" by an explicit two-stage Runge-Rutta scheme solving three ordinary differential equations:

$$\partial_t \mathbf{s}_c(t) = \mathbf{L}_c \tag{A2}$$

       Where, $\mathbf{L}_c = [L_c^0, L_c^{1x}, L_c^{1y}]^T$ includes the respective components of the local discrete spatial DG2 operators, to

update each of the coefficients in $\mathbf{s}_c = [s_c^0(t), s_c^{1x}(t), s_c^{1y}(t)]^T$. The operators in $\mathbf{L}_c$ were already designed to incorporate robust treatments of the bed and friction source terms and of moving wet-dry fronts (Kesserwani & Sharifian, 2020; Shaw et al., 2021).

       The MWDG2 algorithm involves the MRA procedure to decompose, analyse and assemble the shape coefficients $\mathbf{s_c}$, $s \in \{h, hu, hv, z\}$, to produce a non-uniform grid over which the DG2 solver updates are applied (Eq. A2). The MWDG2

algorithm was substantially redesigned to enable efficient parallelisation on the GPU (Chowdhury et al., 2023; Kesserwani & Sharifian, 2023). An overview of the GPU parallelised MWDG2 algorithm (GPU-MWDG2) is provided next.

       In the CUDA programming model for parallelisation the GPU (NVIDIA, 2023), instructions are executed in parallel by workers called "threads", and a group of 32 consecutive threads that operate in lockstep is called a "warp". To devise an efficiently parallelised GPU-MWDG2 code, coalesced memory access, occurring when threads in a warp access contiguous

memory locations, should be maximised, and warp divergence, occurring when threads within a single warp perform different instructions, should be minimised. To achieve these requirements in the GPU-MWDG2 code, the implementation of the MRA procedure had to be reformulated so as to ensure the DG2 solver updates are applicable cell-wise, like with GPU-DG2 (Shaw et al., 2021).

**A.1. MRA procedure**

The MRA procedure must start from a *square uniform grid* at the finest resolution, namely at the given DEM resolution, that is taken to have a maximum refinement level, $L$. This finest grid contains $2^L \times 2^L$ cells, on which the shape coefficients $\mathbf{s}_c^{(L)}$ are initialised. From the finest grid, the MRA procedure can be applied to build a hierarchy of grids of successively coarser resolution, at levels $n = L - 1, \ldots, 1, 0$, with $2^n \times 2^n$ cells. Using the "encoding" operation, the shape coefficients, $\mathbf{s}_c^{(n)}$, and

their associated "details", $\mathbf{d}_{c,\theta}^{(n)} = [d_{c,\theta}^{0,(n)}, d_{c,\theta}^{1x,(n)}, d_{c,\theta}^{1y,(n)}]^T$, $\theta = \alpha, \beta, \gamma$, can be produced on the "parent" cells of the coarser resolution grids, at level $n$, from the shape coefficients $\mathbf{s}_{[0]}^{(n+1)}, \mathbf{s}_{[1]}^{(n+1)}, \mathbf{s}_{[2]}^{(n+1)}$ and $\mathbf{s}_{[3]}^{(n+1)}$ of the four "children" cells at the finer resolution grids, at level $n + 1$ (Eq. 30 in Kesserwani & Sharifian, 2020). However, with the GPU-MWDG2 solver, $\mathbf{s}_c^{(n)}$ and $\mathbf{d}_{c,\theta}^{(n)}$ are stored in arrays in GPU memory that are indexed using Z-order curves (Chowdhury et al., 2023), as exemplified




in Figure A1a for a simplistic case with $L = 2$. With this indexing, coalesced memory access was ensured with the GPU-MWDG2 solver because the shape vectors, $s_{[0]}^{(n+1)}$, $s_{[1]}^{(n+1)}$, $s_{[2]}^{(n+1)}$ and $s_{[3]}^{(n+1)}$ reside in adjacent memory locations when used to produce $s_c^{(n)}$ and $d_{c,\theta}^{(n)}$ (see Figure A1b).

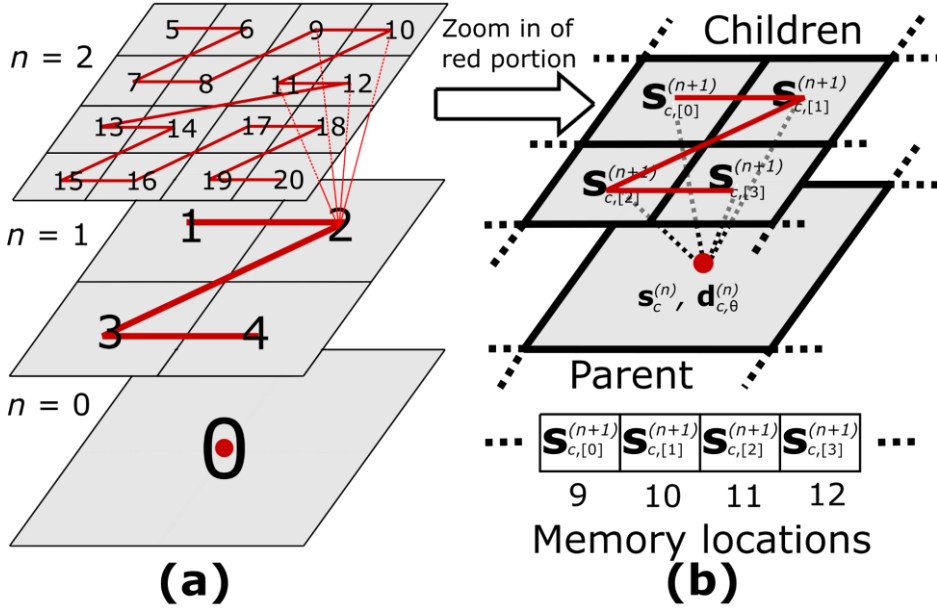

**Figure A1**: Indexing and storage for the MRA procedure on the GPU. Left panel shows a hierarchy of grids across which cells are indexed along the Z-order curve. Right panel shows how four "children" cells at resolution level $n + 1$, and their "parent" cell at resolution level $n$, noting that the shape coefficients at the "children" cells are stored in adjacent GPU memory locations.

While encoding, the magnitude of all the details $d_{c,\theta}^{(n)}$ is analysed in order to identify significant details (Kesserwani & Sharifian, 2020), which results in a tree-like structure of significant details (Figure A2a). The MRA procedure then refers to this tree to generate the non-uniform grid via the "decoding" operation. Decoding is applied within the hierarchy of grids, starting from the coarsest resolution grid until reaching the "leaf" cells, where significant details belong (i.e. a branch of the tree terminates, see Figure A2a where leaf cells are coloured). After decoding, the shape vectors, $s_{[0]}^{(n+1)}$, $s_{[1]}^{(n+1)}$, $s_{[2]}^{(n+1)}$ and $s_{[3]}^{(n+1)}$ at the leaf cells are restored on the non-uniform grid (Eq. 31 in Kesserwani & Sharifian, 2020) to be updated in time.

With the GPU-MWDG2 solver, decoding must be performed using a parallel tree traversal algorithm (PTT) to minimise warp divergence (Chowdhury et al., 2023). To do so, the PTT starts by launching as many threads $t_n$ as the number of cells on the finest resolution grid; for example, 16 threads $t_0$ to $t_{15}$ for traversing the tree in Figure A2a. Each thread independently traverses the tree starting from the cell on the coarsest resolution grid until it reaches its leaf cell $c$ and records its index (Figures A2b and A2c).



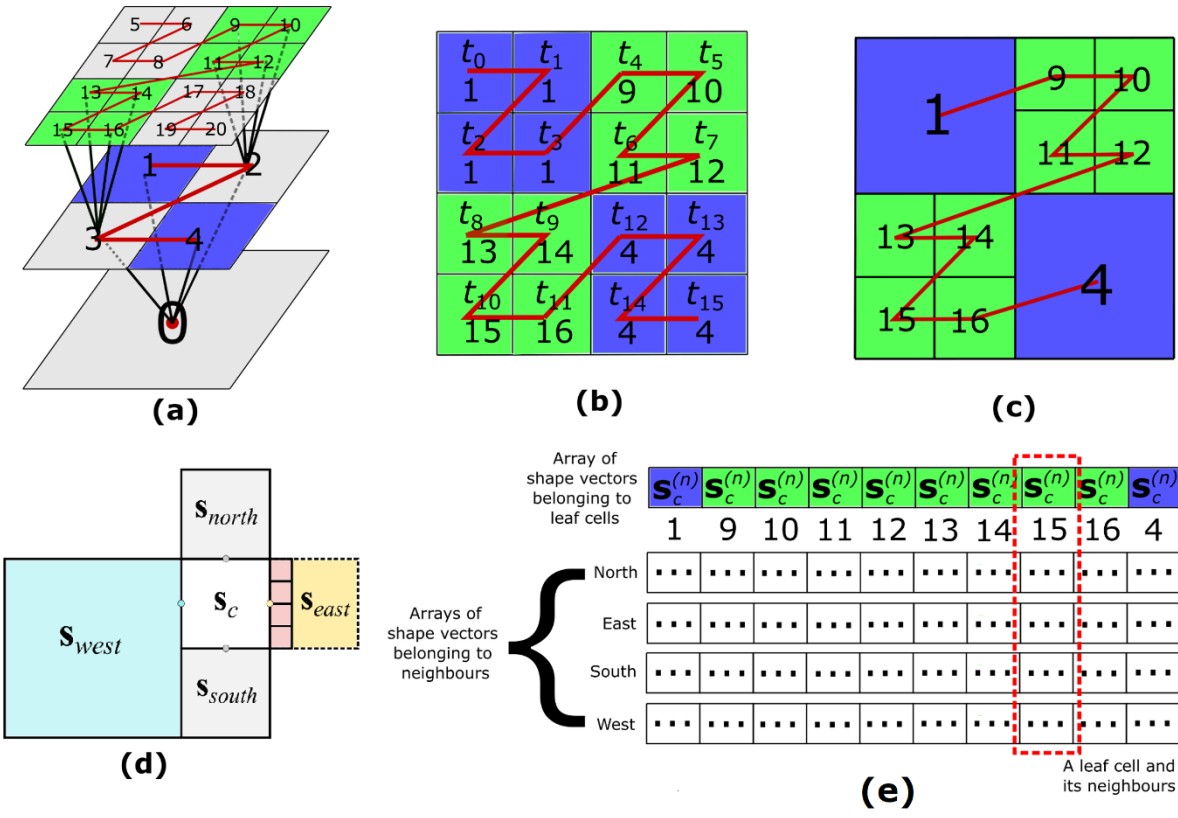

**Figure A2**: Parallel tree traversal (PTT) and neighbour finding. (a) The tree-like structure obtained after flagging significant details during the process of encoding; (b) The leaf cells where the tree terminates (highlighted in green and blue); (c) Leaf cells are assembly into the non-uniform grid; (e) Possible scenarios of neighbouring cells to leaf cell $c$; and (e) leaf and neighbour cells storage as arrays in GPU memory.

Figure A2b shows the indices of the leaf cells identified by each thread once PTT is complete. Since the PTT started with 16 threads and there are fewer leaf cells than the threads, many of threads ended up identifying the same index of the leaf cell (e.g., $t_0$ to $t_3$ identified the leaf cell with index 1 and $t_{12}$ to $t_{15}$ identified the leaf cell with index 4). Threads with duplicate indices are re-used, alongside the other threads, to search and record the indices of east, west, north, and south neighbouring cells of each leaf cell by making each thread look up, down, left, and right. For example, $t_0$ to $t_3$ of the leaf cell with index 1 will identify the east neighbour cells 9 and 11 (Figure A2c). Since the DG2 solver updates on the leaf cell with index 1 requires the (shape coefficients of the) east neighbour cells (shaded red, Figure A2d) to be at the same resolution level as the coarser leaf cell, the PPT will instead record the index of coarsened east neighbour cells (yellow shaded, Figure A2d). For any other scenario (e.g., west, north, or south neighbour cells in Figure A2d), the actual indices and shape coefficients are recorded by the PTT. After recording the indices and shape coefficients, $s_c^{(n)}$, which are to unique each leaf





cell $c$, and its neighbours, $s_{north}^{(n)}$, $s_{south}^{(n)}$, $s_{east}^{(n)}$ and $s_{west}^{(n)}$, they are stored on the non-uniform grid (Figure A2c). In particular, shape coefficients for the leaf cells, $s_c^{(n)}$, are stored in a separate arrays in GPU memory, and separate arrays are also used to do the same for the shape coefficients of their neighbour cells, $s_{north}^{(n)}$, $s_{south}^{(n)}$, $s_{east}^{(n)}$ and $s_{west}^{(n)}$ (see Figure A2e). With this cell-wise storage of indices and shape coefficients, the DG2 solver updates, Eq. A2, can be applied in a straightforward manner.


## A.2 DG2 solver update on the non-uniform grid

On the non-uniform grid, the DG2 solver updates, Eq. A2, are applied to update the shape coefficients $s_c^{(n)}$ by half a timestep over the first Runge-Kutta time stage. After this, another re-encoding step must be applied to the update shape coefficients $s_c^{(n)}$ so that the stored shape coefficients of the four neighbours, $s_{north}^{(n)}$, $s_{south}^{(n)}$, $s_{east}^{(n)}$ and $s_{west}^{(n)}$, are also lifted by half a

timestep. Now, the shape coefficients $s_c^{(n)}$ can be updated by a full timestep by completing the second of Runge-Kutta time stage.

## Appendix B: Step-by-step instructions for running the "Monai valley" example

This Appendix shows how to run a simulation of the "Monai valley" example (Sect. 3.1) using the GPU-MWDG2 solver step-by-step. To use the GPU-MWDG2 solver, the LISFLOOD-FP source code has to be downloaded (LISFLOOD-FP

developers, 2024; https://zenodo.org/doi/10.5281/zenodo.4073010), and then an executable file that can be run has to be built, either on Windows or Linux. To build the executable file on Windows, 1) the LISFLOOD-FP folder should be opened in Visual Studio, 2) either the `x64-Debug` or `x64-Release` option should be selected from the dropdown menu near the toolbar at the top; and, 3) the `Build > Rebuild All` option should be clicked. If the `x64-Debug` option was selected, the executable file, named `lisflood.exe` and containing the GPU-MWDG2 solver, should be built and located in the folder at

`LISFLOOD-FP\out\build\x64-Debug` or similar (or `LISFLOOD-FP\out\build\x64-Release` if the `x64-Release` option was selected). To build on Linux, the steps are 1) navigating to the LISFLOOD-FP directory, 2) running `cmake -S . -B build` in the terminal; and, 3) running `cmake --build build`. The executable file, named `lisflood`, should be built and located in the `LISFLOOD-FP/build` directory.

     After the executable file has been built, it can be run in order to run simulations of the "Monai valley" example

using the GPU-MWDG2 solver. Before running the simulation, several input files must be prepared, which are listed in Table B.1. To prepare the input files, a number of Python scripts should be used that are available in the `monai` folder uploaded alongside the input files made publicly available online for reproducing the results and simulations reported in this paper (Chowdhury, 2024; https://doi.org/10.5281/zenodo.13909072). The usage of these Python scripts is as follows.


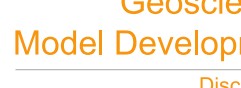
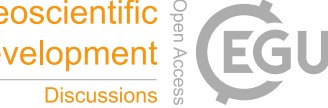

**Table B.1**: Input files needed to run simulations of the Monai valley example using the GPU-MWDG2 solver.

| Input file | File name | Description |
|---|---|---|
| Digital elevation model | `monai.dem` | ASCII raster file containing the numerical values of the bathymetric elevation pixel-by-pixel. |
| Initial flow conditions | `monai.start` | ASCII raster file containing the numerical values of the initial water depth and discharge pixel-by-pixel. |
| Boundary conditions | `monai.bci` | Text file specifying where boundary conditions are enforced and what type (fixed versus time-varying). |
| Time series at boundaries | `monai.bdy` | Text file containing time series in case time-varying boundary conditions and/or point sources have been specified in the `.bci` file. |
| Stage locations | `monai.stage` | Text file containing the locations of virtual stage points where simulated time histories of the water depth are recorded. |
| Parameter file | `monai.par` | Text file containing parameters to access various solver and simulation features. |

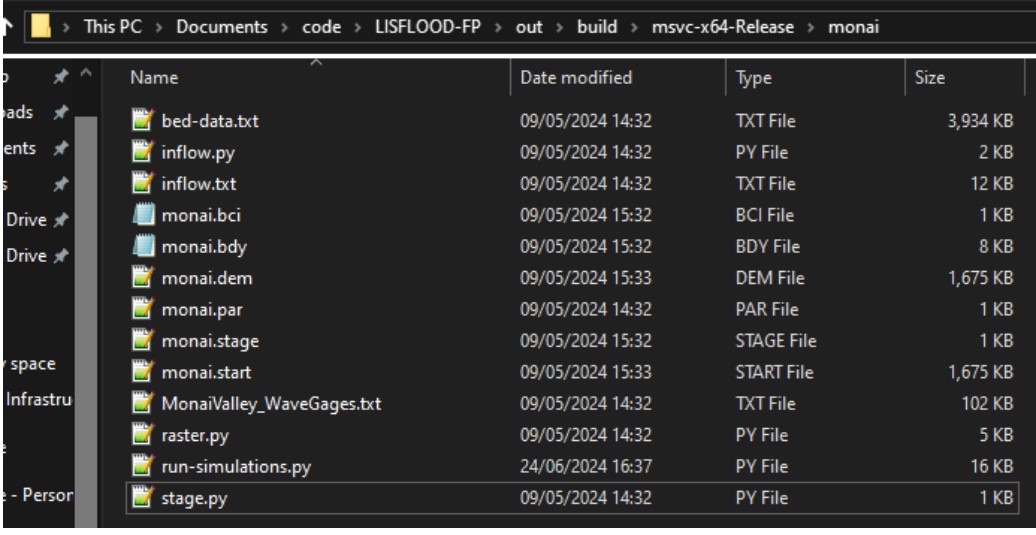

**Figure B.1**: Input files prepared for the "Monai valley" example after running the Python scripts available in the `monai` folder.

To prepare the input files for the Monai valley simulation using the Python scripts, 1) the `monai` folder should be copied to the same location as the `lisflood.exe` executable file, e.g. `LISFLOOD-FP\out\build\x64-Release` if on Windows or `LISFLOOD-FP/build` if on Linux, 2) the `monai` folder should be navigated to, 3) the `monai.stage` file should be generated by typing and running `python stage.py` in a command prompt, 4) the `monai.dem` and `monai.start` raster files should be generated by running `python raster.py`, 5) the `monai.bci` and `monai.bdy` files





should be generated by running `python inflow.py`, 6) the parameter file `monai.par` should be prepared as shown in Figure 2; and, 7) the simulation should be run by typing and running `..\lisflood.exe monai.par` in a command prompt. Following steps (1) to (7) should result in the files shown in Figure B.1. Steps (1) to (7) can be performed as a fully

automated process by running `python run-simulations.py`: this Python script will automatically prepare the input files (steps 3 to 6), run several simulations (step 7), and postprocess the results. Note that if `run-simulations.py` is run inside the downloaded `monai` folder before running any simulations, and with the `self.run()` function in the Python file commented out, it will reproduce the results in Sect. 3.1 (i.e. it will generate Figures 7 and 8 and the data for Table 3).