# Peer review of "multiscale flood simulation"

_Geoscientific Model Development, 2024_

## Author Comment (AC1)

**Reply letter to reviewers**

Dear Editor and Reviewers,

We thank the Editor for coordinating the review process. We are also very grateful to the Reviewers for taking the time to review our manuscript. We found their comments helpful for improving the depth of our manuscript, and we have revised it in order to address the comments. Below, the comments are included for reference in italics, while our replies are in boxes, and screenshot(s) of the relevant revision(s) are given below our replies.

Yours sincerely, Alovya Chowdhury, Georges Kesserwani

**Reviewer 1**

In the manuscript titled "LISFLOOD-FP 8.2: GPU-accelerated multiwavelet discontinuous Galerkin solver with dynamic resolution adaptivity for rapid, multiscale flood simulation", the authors develop the new LISFLOOD-FP 8.2 version integrates GPU parallelised dynamic and the DG2 solver (GPU-MWDG2) to simulate tsunami-induced flooding. It shows progressively larger speedups over the GPU-DG2 simulations from  $L \ge 10$ . There are still certain changes and clarifications that the authors should address prior to publication. There are still certain changes and clarifications that the authors should address prior to publication. I believe that the manuscript can be accepted for publication by the GMD after minor revision. Below, I have some general comments for the authors.

General comments:

The paper describes an innovative GPU-MWDG2 solver, so section 2.1 is the core of the calculation method in this paper. To help readers understand the detailed clarification on certain algorithmic. It is recommended that the authors provide some key portions of the code or pseudo-code as appendix. This is merely a suggestion and does not affect the validity of the paper's arguments.

Key portions of the code, in particular pseudocode and descriptions of the optimised CUDA kernels of the encoding process and the parallel tree traversal (PTT), have now been included in Appendix A.

2023), as exemplified in Figure A1a for a simplistic case with L = 2. Hence,  $s_{[0]}^{(n+1)}$ ,  $s_{[1]}^{(n+1)}$ ,  $s_{[2]}^{(n+1)}$  and  $s_{[3]}^{(n+1)}$  all reside in

adjacent memory locations when used to produce  $s_c^{(n)}$  and  $d_{c,\theta}^{(n)}$  (see Figure A1b) and can be accessed using vectorised

920 float4 instructions (see lines 7 to 22 of Algorithm 1) to ensure coalesced memory access. The produced  $s_c^{(n)}$  can then be trivially stored in the appropriate index of the array (see lines 25 to 27 of Algorithm 1), again in a coalesced manner due to the self-similar nature of the Z-order curve between refinement levels *n* and n + 1.

|     | 01                                                                          | global                                                                                                                                                                                                                                                                                                                                                                                                                                                                         |
|-----|-----------------------------------------------------------------------------|--------------------------------------------------------------------------------------------------------------------------------------------------------------------------------------------------------------------------------------------------------------------------------------------------------------------------------------------------------------------------------------------------------------------------------------------------------------------------------|
| 930 | 02 V0                                                                       | Did encode flow kernel_mw                                                                                                                                                                                                                                                                                                                                                                                                                                                      |
|     | 03 (                                                                        |                                                                                                                                                                                                                                                                                                                                                                                                                                                                                |
|     | 04                                                                          | // Inputs                                                                                                                                                                                                                                                                                                                                                                                                                                                                      |
|     | 05)                                                                         |                                                                                                                                                                                                                                                                                                                                                                                                                                                                                |
|     | 06 {                                                                        |                                                                                                                                                                                                                                                                                                                                                                                                                                                                                |
| 935 | 07                                                                          | <pre>// Compute Z order indices based on thread index</pre>                                                                                                                                                                                                                                                                                                                                                                                                                    |
|     | 08                                                                          | HierarchyIndex idx = blockIdx.x * blockDim.x + threadIdx.x;                                                                                                                                                                                                                                                                                                                                                                                                                    |
|     | 09                                                                          | HierarchyIndex curr_lvl_idx = get_lvl_idx(level);                                                                                                                                                                                                                                                                                                                                                                                                                              |
|     | 10                                                                          | HierarchyIndex next_lvl_idx = get_lvl_idx(level + 1);                                                                                                                                                                                                                                                                                                                                                                                                                          |
|     | 11                                                                          | HierarchyIndex parent_idx = curr_lvl_idx + idx;                                                                                                                                                                                                                                                                                                                                                                                                                                |
| 940 | 12                                                                          | HierarchyIndex child_idx = next_lvl_idx + 4 * (parent_idx - curr_lvl_idx);                                                                                                                                                                                                                                                                                                                                                                                                     |
|     | 13                                                                          |                                                                                                                                                                                                                                                                                                                                                                                                                                                                                |
|     | 14                                                                          | <pre>// Load 4 child scale coefficients in a coalesced manner using vector loads (float4)</pre>                                                                                                                                                                                                                                                                                                                                                                                |
|     | 15                                                                          | load_children_vector                                                                                                                                                                                                                                                                                                                                                                                                                                                           |
|     | 16                                                                          | (                                                                                                                                                                                                                                                                                                                                                                                                                                                                              |
| 945 | 17                                                                          | children,                                                                                                                                                                                                                                                                                                                                                                                                                                                                      |
|     | 18                                                                          | d_scale_coeffs.eta0,                                                                                                                                                                                                                                                                                                                                                                                                                                                           |
|     | 19                                                                          | d_scale_coeffs.eta1x,                                                                                                                                                                                                                                                                                                                                                                                                                                                          |
|     | 20                                                                          | d_scale_coeffs.eta1y,                                                                                                                                                                                                                                                                                                                                                                                                                                                          |
|     | 21                                                                          | child_idx                                                                                                                                                                                                                                                                                                                                                                                                                                                                      |
| 950 | 22                                                                          | );                                                                                                                                                                                                                                                                                                                                                                                                                                                                             |
|     | 23                                                                          |                                                                                                                                                                                                                                                                                                                                                                                                                                                                                |
|     | 24                                                                          | <pre>// Use encoding equation to compute and store parent scale coefficient</pre>                                                                                                                                                                                                                                                                                                                                                                                              |
|     | 25                                                                          | d_scale_coeffs.eta0[parent_idx] = encode_scale_0 (children);                                                                                                                                                                                                                                                                                                                                                                                                                   |
|     | 26                                                                          | <pre>d_scale_coeffs.eta1x[parent_idx] = encode_scale_1x(children);</pre>                                                                                                                                                                                                                                                                                                                                                                                                       |
| 955 | 27                                                                          | d_scale_coeffs.eta1y[parent_idx] = encode_scale_1y(children);                                                                                                                                                                                                                                                                                                                                                                                                                  |
|     | 28 }                                                                        |                                                                                                                                                                                                                                                                                                                                                                                                                                                                                |
| 960 | Algorithm 1: O
Z-order indices
vectorised float4
similar nature of | ptimised CUDA kernel for performing the encoding operation in parallel. Coalesced memory access is ensured because the computed per thread point to adjacent memory locations in the array that is used to $s_c^{(n)}$ and $d_{c,\theta}^{(n)}$ . As seen in line 15, 4 loads can be used to load the children when computing the parent. Storing the parent is trivially coalesced due to the self-
f-Z-order indexing across refinement levels n and n + 1. |
|     | With                                                                        | the GPU-MWDG2 solver, decoding must be performed using a parallel tree traversal algorithm (PTT) to                                                                                                                                                                                                                                                                                                                                                                            |
| 970 | minimise war                                                                | p divergence (Chowdhury et al., 2023). To do so, the PTT starts by launching as many threads $t_n$ as the                                                                                                                                                                                                                                                                                                                                                                      |
|     | number of cel                                                               | ls on the finest resolution grid; for example, 16 threads $t_0$ to $t_{15}$ for traversing the tree in Figure A2a. Each                                                                                                                                                                                                                                                                                                                                                        |
|     | thread indeper                                                              | idently traverses the tree starting from the cell on the coarsest resolution grid until it reaches its leaf cell c, i.e.                                                                                                                                                                                                                                                                                                                                                       |
|     | until it reaches                                                            | s either a detail that is no longer significant (line 14 of Algorithm 2) or the finest refinement level (line 28 of                                                                                                                                                                                                                                                                                                                                                            |
|     | Algorithm 2)                                                                | Once a thread reaches a leaf cell, it records the Z-order index of that leaf cell (Figures A2b and A2c)                                                                                                                                                                                                                                                                                                                                                                        |
|     |                                                                             |                                                                                                                                                                                                                                                                                                                                                                                                                                                                                |

|    |      | 01 _                                                | global                                                                                                                                                                                                                                                                                                                      |
|----|------|-----------------------------------------------------|-----------------------------------------------------------------------------------------------------------------------------------------------------------------------------------------------------------------------------------------------------------------------------------------------------------------------------|
|    |      | 02 v                                                | oid traverse_tree_of_sig_details                                                                                                                                                                                                                                                                                            |
|    |      | 03 (                                                |                                                                                                                                                                                                                                                                                                                             |
|    | 985  | 04                                                  | // Inputs                                                                                                                                                                                                                                                                                                                   |
|    |      | 05 )                                                |                                                                                                                                                                                                                                                                                                                             |
|    |      | 06 {                                                |                                                                                                                                                                                                                                                                                                                             |
|    |      | 07                                                  | <pre>while (keep_on_traversing)</pre>                                                                                                                                                                                                                                                                                       |
|    |      | 08                                                  | {                                                                                                                                                                                                                                                                                                                           |
|    | 990  | 09                                                  | MortonCode curr_code = ( fine_code >> ( 2 * (solver_params.L - level) ) );                                                                                                                                                                                                                                                  |
|    |      | 10                                                  | HierarchyIndex curr_lvl_idx = get_lvl_idx(level);                                                                                                                                                                                                                                                                           |
|    |      | 11                                                  | HierarchyIndex h_idx = curr_lvl_idx + curr_code;                                                                                                                                                                                                                                                                            |
|    |      | 12                                                  | <pre>bool is_sig = d_sig_details[h_idx];</pre>                                                                                                                                                                                                                                                                              |
|    |      | 13                                                  |                                                                                                                                                                                                                                                                                                                             |
|    | 995  | 14                                                  | <pre>if (!is_sig)</pre>                                                                                                                                                                                                                                                                                                     |
|    |      | 15                                                  | {                                                                                                                                                                                                                                                                                                                           |
|    |      | 16                                                  | // Record Z order index                                                                                                                                                                                                                                                                                                     |
|    |      | 17                                                  | <pre>keep_on_traversing = false;</pre>                                                                                                                                                                                                                                                                                      |
|    |      | 18                                                  | }                                                                                                                                                                                                                                                                                                                           |
|    | 1000 | 19                                                  | else                                                                                                                                                                                                                                                                                                                        |
|    |      | 20                                                  | {                                                                                                                                                                                                                                                                                                                           |
|    |      | 21                                                  | <pre>if (!penultimate_level)</pre>                                                                                                                                                                                                                                                                                          |
|    |      | 22                                                  | {                                                                                                                                                                                                                                                                                                                           |
|    |      | 23                                                  | <pre>keep_on_traversing = true;</pre>                                                                                                                                                                                                                                                                                       |
|    | 1005 | 24                                                  | }                                                                                                                                                                                                                                                                                                                           |
|    |      | 25                                                  | else                                                                                                                                                                                                                                                                                                                        |
|    |      | 26                                                  | {                                                                                                                                                                                                                                                                                                                           |
|    |      | 27                                                  | // Record Z order index                                                                                                                                                                                                                                                                                                     |
|    |      | 28                                                  | <pre>keep_on_traversing = false;</pre>                                                                                                                                                                                                                                                                                      |
|    | 1010 | 29                                                  | }                                                                                                                                                                                                                                                                                                                           |
|    |      | 30                                                  | }                                                                                                                                                                                                                                                                                                                           |
|    |      | 31                                                  | }                                                                                                                                                                                                                                                                                                                           |
|    |      | 32 }                                                |                                                                                                                                                                                                                                                                                                                             |
|    | 1015 | Algorithm 2:
traversing the t
reduced because | Deptimised CUDA kernel for performing parallel tree traversal (PTT) that minimises warp divergence. Threads ke
ee of significant details until they reach either an insignificant detail or the finest refinement level. Warp divergence
threads that end up reaching nearby leaf cells have similar traversal paths. |
| 11 |      |                                                     |                                                                                                                                                                                                                                                                                                                             |

The paper could benefit from a deeper analysis of the trade-offs between accuracy and efficiency when choosing different error thresholds ( $\epsilon$ ). The choice of  $\epsilon$  = 10-4 vs. 10-3 shows efficiency improvements, more discussion is needed on when to choose one over the other in practical scenarios.

More discussion on when to choose one over the other in practical scenarios has been included in the conclusion. This discussion is in addition to other substantial revisions about validating these choices of  $\varepsilon$  throughout Sect. 3. The accuracy and efficiency of dynamic GPU-MWDG2 adaptivity was assessed for four laboratory- and field-scale 670 tsunami-induced flood benchmarks featuring different impact event's complexity and duration (i.e. incorporating either single- or multi-peaked tsunamis and L = 10 or L = 12). GPU-MWDG2 simulation assessments were performed for  $\varepsilon = 10^{-3}$  and  $10^{-4}$ , which were also validated based on accuracy qualification with respect to benchmark-specific measured data. The accuracy assessment consistently confirms that an  $\varepsilon$  between  $10^{-4}$  and  $10^{-3}$  is valid: at  $\varepsilon = 10^{-3}$  GPU-MWDG2 simulations yield water level predictions as accurate as the GPU-DG2 predictions but using  $\varepsilon = 10^{-4}$  can slightly improve velocity-related 675 predictions.

Speedup assessments, based on instantaneous and cumulative metrics, suggested considerable gain when the DEM size and resolution involved L ≥ 10 and for simulation durations that mostly spanned reduced-complexity events (i.e. single-peak tsunamis): As shown in Table 7, for single-peaked tsunamis, when the DEM required L = 10, GPU-MWDG2 was more than 2-fold faster than the GPU-DG2 simulations (i.e. "Monai Valley"), whereas with the DEM requiring L = 12, speedups
of 3.3-to-4.5-fold could be achieved (i.e. "Seaside, Oregon"); in contrast, for multi-peaked tsunamis, GPU-MWDG2 speedups reduced to 1.8-fold for a DEM requiring L = 12 (i.e. "Tauranga Harbour") and to 1.2-fold for a smaller DEM

needing L = 10 (i.e. "Hilo Harbour").

|                  |                |                     |                               |              | GPU-MWDG2 |              |      | GPU-DG2
Runtime |
|------------------|----------------|---------------------|-------------------------------|--------------|-----------|--------------|------|--------------------|
|                  | Tsunami (imj   | pact) event         | L (square
uniform grid)    | Runtime
ε |           | Speedup
E |      |                    |
| Test case        | tend           | Single-wave tsunami |                               | 10-3         | 10-4      | 10-3         | 10-4 |                    |
| Monai Valley     | 22.5 (9000 s*) | Yes                 | $10 \ (2^{10} \times 2^{10})$ | 16 s         | 20 s      | 2.5          | 2.0  | 40 s               |
| Seaside Oregon   | 40 s (33 min*) | Yes                 | $12 (2^{12} \times 2^{12})$   | 3.5 min      | 5.2 min   | 4.5          | 3.3  | 13 min             |
| Tauranga Harbour | 40 hr          | No (three peaks)    | $12(2^{12}\times 2^{12})$     | 7.5 hr       | 8.1 hr    | 1.8          | 1.4  | 11.3 hr            |
| Hilo Harbour     | 6 hr           | No (eleven peaks)   | $10 \ (2^{10} \times 2^{10})$ | 5.3 min      | 5.8 min   | 1.3          | 1.2  | 6.9 min            |

685 \*By accounting for the physical scaling factor of the replica.

In summary, the GPU-MWDG2 solver in LISFLOOD-FP 8.2 consistently accelerates GPU-DG2 simulations of rapid multiscale flooding flows, generally yielding the greatest speedups for simulations needing  $L \ge 10$ —due to its scalability on the GPU which results in larger speedup with increasing *L*—and driven by single-peaked impact events. 690 Choosing  $\varepsilon$  closer to 10-3 would maximise speedup while choosing an  $\varepsilon$  closer 10-4 may be useful to particularly improve the velocity-related predictions. The LISFLOOD-FP 8.2 code is open source (DOI: 10.5281/zenodo.4073010) as well as the

Figure 7 and Figure 10 are comparisons between GPU-MWDG2 and GPU-DG2, there is no description of the dashed line in the figure caption. The comparison between the two in terms of computational performance is not intuitive enough.

Figure captions have been added that explain the dashed line and include more intuitive explanations of the metrics for assessing the computational performance, e.g. in Sect. 3.1 Monai Valley.

375 Figure 7: Monai Valley. Measuring the computational performance of the GPU-MWDG2 and GPU-DG2 simulations using the metrics of Table 1 to assess the speedup afforded by GPU-MWDG2 adaptivity: Ncells, the number of cells in the GPU-MWDG2 non-uniform grid, where the dashed line represents the initial value at the beginning of the simulation; RDG2, the computational effort of performing the DG2 solver updates at given timestep relative to the GPU-DG2 simulation; RMRA, the relative computational effort of performing the MRA process at a given timestep; Sinst, the instantaneous speedup achieved by the GPU-MWDG2 simulation over the GPU-DG2 simulation at a given timestep; At, the simulation timestep; NAt, the number of timesteps taken to reach a given simulation time; CDG2, the cumulative computational effort of performing the DG2 solver updates up to a given simulation time; CMRA, the cumulative computational effort of GPU-MWDG2 over GPU-DG2 up to a given simulation time.

The paper touches on the scalability of the solver but could provide a more forward-looking discussion of future applications. For example, how would this solver perform in larger simulations involving urban flooding, river flooding that require coupling with other